# [1]LSCE-FFNN-v1: A two-step neural network model for the [2]reconstruction of surface ocean pCO$_2$ over the Global [3]Ocean.

[5]Anna Denvil-Sommer[1], Marion Gehlen[1], Mathieu Vrac[1], Carlos Mejia[2]

[6][1]Laboratoire des Sciences du Climat et de l'Environnement (LSCE), Institut Pierre Simon Laplace (IPSL), [7]CNRS/CEA/UVSQ/Univ. Paris-Saclay, Orme des Merisiers, Gif-Sur-Yvette, 91191, France

[8][2]Sorbonne Université, CNRS, IRD, MNHN, Institut Pierre Simon Laplace (IPSL), Paris, 75005, France

[10]*Correspondence to*: Anna Denvil-Sommer (anna.sommer.lab@gmail.com)

[12]**Abstract.**

[13]A new Feed-Forward Neural Network (FFNN) model is presented to reconstruct surface ocean partial [14]pressure of carbon dioxide (pCO$_2$) over the global ocean. The model consists of two steps: (1) [15]reconstruction of pCO$_2$ climatology and (2) reconstruction of pCO$_2$ anomalies with respect to the [16]climatology. For the first step, a gridded climatology was used as the target, along with sea surface salinity [17]and temperature (SSS and SST), sea surface height (SSH), chlorophyll *a* (Chl), mixed layer depth (MLD), [18]as well as latitude and longitude as predictors. For the second step, data from the Surface Ocean CO$_2$ Atlas [19](SOCAT) provided the target. The same set of predictors was used during step 2 augmented by their [20]anomalies. During each step, the FFNN model reconstructs the non-linear relationships between pCO$_2$ and [21]the ocean predictors. It provides monthly surface ocean pCO$_2$ distributions on a 1°x1° grid for the period [22]2001-2016. Global ocean pCO$_2$ was reconstructed with a satisfying accuracy compared to independent [23]observational data from SOCAT. However, errors are larger in regions with poor data coverage (e.g. Indian [24]Ocean, Southern Ocean, subpolar Pacific). The model captured the strong interannual variability of surface [25]ocean pCO$_2$ with reasonable skills over the Equatorial Pacific associated with ENSO (El Niño Southern [26]Oscillation). Our model was compared to three pCO$_2$ mapping methods that participated in the Surface [27]Ocean pCO$_2$ Mapping intercomparison (SOCOM) initiative. We found a good agreement in seasonal and [28]interannual variability between the models over the global ocean. However, important differences still exist [29]at the regional scale, especially in the Southern Hemisphere and in particular, the Southern Pacific and the [30]Indian Ocean, as these regions suffer from poor data-coverage. Large regional uncertainties in [31]reconstructed surface ocean pCO$_2$ and sea-air CO$_2$ fluxes have a strong influence on global estimates of [32]CO$_2$ fluxes and trends.

33

**1. Introduction.**

The global ocean is a major sink of excess $CO_2$ emitted to the atmosphere since the beginning of the industrial revolution. In 2011, the best estimate of the ocean inventory of anthropogenic carbon ($C_{ant}$) amounts to 155 ± 30 PgC or 28% of cumulated total $CO_2$ emissions attributed to human activities since 1750 (Ciais et al., 2013). Between 2000 and 2009, the yearly average ocean $C_{ant}$ uptake was 2.3 ± 0.7 PgC $yr^{-1}$ (Ciais et al., 2013). However, these global estimates hide substantial regional and inter-annual fluctuations (Rödenbeck et al., 2015), which need to be quantified in order to track the evolution of the Earth's carbon budget (e.g. Le Quéré et al., 2018).

Until recently, most estimates of inter-annual sea-air $CO_2$ flux variability were based on atmospheric inversions (Peylin et al., 2005, 2013; Rödenbeck et al., 2005) or global ocean circulation models (Orr et al., 2001; Aumont and Bopp, 2006; Le Quéré et al., 2010). However, models tend to underestimate the variability of sea-air $CO_2$ fluxes (Le Quéré et al., 2003), while atmospheric inversions suffer from a still sparse network of atmospheric $CO_2$ measurements (Peylin et al., 2013). These approaches are increasingly complemented by data-based techniques relying on *in situ* measurements of $CO_2$ fugacity or partial pressure (e.g. Takahashi et al., 2002, 2009; Nakaoka et al., 2013; Schuster et al., 2013; Landschützer et al., 2013, 2016; Rödenbeck et al., 2014, 2015; Bitting et al., 2018;   Fay et al., 2014;). These techniques rely on a variety of data-interpolation approaches developed to provide estimates in time and space of surface ocean $pCO_2$ (Rödenbeck et al., 2015) such as statistical interpolation, linear and non-linear regressions, or model-based regressions or tuning (Rödenbeck et al., 2014, 2015). These methods, their advantages and disadvantages are compared and discussed in Rödenbeck et al. (2015). This intercomparison did not allow identifying a single optimal technique but rather pleaded in favour of exploiting the ensemble of methods.

Artificial neural networks (ANN) have been widely used to reconstruct surface ocean $pCO_2$ (open ocean: Lefèvre et al., 2005; Friedrich and Oschlies, 2009b; Telszewski et al., 2009; Landschützer et al., 2013; Nakaoka et al., 2013; Zeng et al. 2014; Bitting et al., 2018; coastal region: Laruelle et al., 2017). ANN fill the spatial and temporal gaps based on calibrated non-linear statistical relationships between $pCO_2$ and its oceanic and atmospheric drivers. The existing products usually present monthly fields with a 1º×1º spatial resolution and capture a large part of temporal-spatial variability. Methods based on ANN are able to represent the relationships between pCO2 and a variety of predictor combinations, but they are sensitive to the number of data used in the training algorithm and can generate artificial variability in regions with sparse data coverage (Bishop, 2006).

This study proposes an alternative implementation of a neural network applied to the reconstruction of surface ocean $pCO_2$ over the period 2001-2016. It belongs to the category of Feed Forward Neural

Networks (FFNN) and consists of a two-step approach: (1) the reconstruction of monthly climatologies of

global surface ocean pCO$_2$ based on data from Takahashi et al. (2009), and (2) the reconstruction of

monthly anomalies (with respect to the monthly climatologies) on a 1°x1° grid exploiting the Surface

Ocean CO$_2$ Atlas (SOCAT) (Bakker et al., 2016). The model is easily applied to the global ocean without

any boundaries between the ocean basins or regions. However, as mentioned before, it is still sensitive to

the observational coverage. This limitation is partly overcome by the two-step approach as the

reconstruction of monthly climatologies draws on a global ocean gridded climatology (Takahashi et al.,

2009), thereby keeping FFNN output close to realistic values. Furthermore, the reconstruction of monthly

climatologies during the first step allows taking into account a potential change in seasonal cycle in

response to climate change when applied to time slices or to model output providing the drivers, but no

carbon cycle variables.

The remainder of this paper is structured as follows: section 2 introduces datasets used during this study

and describes the neural network; section 3 presents results for its validation and qualification, as well as a

comparison to three mapping methods part of the Surface Ocean pCO$_2$ Mapping intercomparison

(SOCOM) exercise (Rödenbeck et al., 2015). Results and perspectives are summarized in the last section.

**2. Data and method.**

2.1. Data.

The standard set of variables known to represent physical, chemical and biological drivers of surface ocean

pCO$_2$ – mean state and variability – (Takahashi et al., 2009; Landschützer et al., 2013) were used as input

variables (or predictors) for training the FFNN algorithm. These are sea surface salinity (SSS), sea surface

temperature (SST), mixed layer depth (MLD), chlorophyll *a* concentration (CHL), atmospheric CO$_2$ mole

fraction ($x$CO$_{2,atm}$). Based on Rodgers et al. (2009) who reported a strong correlation between natural

variations in dissolved inorganic carbon (DIC) and sea surface height (SSH), SSH was added as a new

driver to this list. First tests suggested that the inclusion of SSH does not significantly improve the accuracy

of reconstructed pCO$_2$ at global scale. At basin and regional scale, however, adding SSH improves the

spatial pattern of reconstructed pCO$_2$ and the accuracy of our method.

For the first step, the reconstruction of monthly climatologies, the Takahashi et al. (2009) monthly pCO$_2$

gridded climatology (1°x1°) was used as the target. The original climatology was constructed by an

advection-based interpolation method on a 4°x5° grid. It was interpolated on the 1°x1° SOCAT grid which

is also the resolution of the final output for the FFNN.

For the second step, the target is provided by the observational database SOCAT v5 (Bakker et al., 2016).

We used a gridded version of this dataset that was derived by combining all SOCAT data collected within a

1°x1° box during a specific month. SOCAT v5 represents global observations of sea surface fugacity of CO$_2$

($f$CO$_2$) over the period 1970 to 2016. It includes data from moorings, ships and drifters. These data are

106distributed irregularly over the global ocean with 188274 gridded measurements over the Northern

107Hemisphere and 76065 over the Southern Hemisphere. In order to ensure a satisfying spatial and temporal

108data coverage, we limited the reconstruction to the period 2001-2016, which represents ~77% of the

109database (Fig. 1(a)).

110The following formula is used to convert $fCO_2$ to $pCO_2$ (Körtzinger et al., 1999):

$$111 \quad fCO_2 = pCO_2 \exp\left( p\,\frac{B+2\delta}{RT} \right) , \quad (1)$$

112where $fCO_2$ and $pCO_2$ are in µatm, p is the total pressure (Pa), R=8.314 JK$^{-1}$ is the gas constant, T is the

113absolute temperature (K). Parameter B (m$^3$mol$^{-1}$) is estimated as: B = (-1636.75 + 12.0408 T – 3.27957 *

114$10^{-2}$ T$^2$ + 3.16528 * $10^{-5}$ T$^3$) $10^{-6}$ . The parameter $\delta$ is the cross virial coefficient (m$^3$mol$^{-1}$): $\delta$ = (57.7 –

1150.118T) $10^{-6}$. The total pressure is from the Jena database (6h, 5°x5°) based on the NCEP reanalysis (Kalnay

116et al., 1996) (http://www.bgc-jena.mpg.de/CarboScope/?ID=s).

118Monthly global reprocessed products of physical variables from ARMOR3D L4 distributed through the

119Copernicus Marine Environment Monitoring Service (CMEMS) (0.25°x0.25°)

120(http://marine.copernicus.eu/services-portfolio/access-to-products/?

121option=com_csw&view=details&product_id=MULTIOBS_GLO_PHY_REP_015_002) were used for SSS,

122SST and SSH (Guinehut et al., 2012). The GlobColour project provided monthly CHL distributions at 1°x1°

123resolution (http://www.globcolour.info/products_description.html). For MLD, daily data from the

124"Estimating the Circulation and Climate of the Ocean" (ECCO2) project Phase II (Cube 92), at 0.25°x0.25°

125resolution (Menemenlis et al., 2008) were used. For $xCO_2$ atmospheric, the 6h data from Jena $CO_2$

126inversion s76_v4.1 on a 5°x5° grid were selected (http://www.bgc-jena.mpg.de/CarboScope/?ID=s). Finally,

127an ice mask based on daily "Operational Sea Surface Temperature and Sea Ice Analysis" (OSTIA) with a

128gridded 0.05°x0.05° resolution (Donlon et al., 2011) was applied.

129MLD and CHL were log-transformed before their use in the FFNN algorithm because of their skewed

130distribution. In regions with no CHL data (high latitudes in winter) log(CHL) = 0 was applied. It does not

131introduce discontinuities since log(CHL) is close to zero in the adjacent region.

133All data were averaged or interpolated on a 1°x1° grid and, depending on the resolution of the dataset,

134averaged over the month. It is worth noting that all datasets have to be normalized (i.e. centered to zero-

135mean and reduced to unit standard deviation) before their use in the FFNN algorithm, for example:

$$136 \quad SSS_n = \frac{SSS - \overline{SSS}}{std\,(SSS)} .$$

137Normalization ensures that all predictors fall within a comparable range and therefore avoids giving more

138weight to predictors with large variability ranges (Kallache et al., 2011).

139As surface ocean $pCO_2$ also varies spatially, geographical positions (lat, lon) after conversion to radians

were included as predictors. In order to normalize (lat, lon) the following transformation is proposed:

$$lat_n = \sin\left(lat * \pi / 180^0\right)$$

$$lon_{n,1} = \sin\left(lon * \pi / 180^0\right)$$

$$lon_{n,2} = \cos\left(lon * \pi / 180^0\right)$$

Two functions *sin* and *cos* for longitudes are used to preserve its periodical 0 to 360 degrees behavior and
thus to consider the difference of positions before and after the 0º longitude. For step 2, data required for
training were co-located at the SOCAT data positions that are used as a target for the FFNN model. Details
are provided in the next section.

2.2. Method.

a) Network configuration and evaluation protocol

In this work, we use Keras, a high-level neural network Python library ("Keras: The Python Deep Learning
library", Chollet, 2015; https://keras.io) to build and train the FFNN models. The identification of an
optimal configuration is the first step in the FFNN model building. This includes: the choice of number and
size of hidden layers (i.e., intermediate layers between input and output layers), connection type, activation
functions, loss function and optimization algorithm, as well as the learning rate and other low-level
parameters. Based on a series of tests and their statistical results (RMSE, correlation, bias) a hyperbolic
tangent was chosen as an activation function for neurons in hidden layers, and a linear function for the
output layer. As optimization algorithm, the mini-batch gradient descent or RMSprop was used (adaptive
learning rates for each weight, Chollet, 2015; Hinton et al., 2012). The number of layers and neurons
depends on the problem. For totally connected layers (i.e., a neuron in a hidden layer is connected to all
neurons in the precedent layer and connects all neurons in the next one), that is the case here, it is enough to
have only one single hidden layer but two or more can help the approximation of complex functions (or
complex relationships between the input and the output of the problem).

The number of the FFNN layers and the number of neurons depends on one side on the complexity of the
problem: the more layers and neurons, the better the accuracy of the output. However, the size also depends
on the number of patterns (data) used for training. The empirical rule advises to have a factor of 10 between
the number of patterns (data) and the number of connections, or weights to adjust (in line with Amari et al.
(1997), we use a factor of 10 that necessitates a cross-validation to avoid overfitting). This limits the size,
the number of parameters and incidentally the number of neurons, of the FFNN. This empirical rule was
followed in this study.

175(1) Step 1: reconstruction of monthly climatologies

176FFNN reconstructs a normalized monthly surface ocean $pCO_2$ climatology as a nonlinear function of

177normalized SSS, SST, SSH, Chl, MLD climatologies and geographical position (longitude, latitude):

$$pCO_{2,n} = \left( SSS_n, SST_n, SSH_n, Chl_n, MLD_n, lon_n, lat_n \right) \quad (2)$$

179Surface ocean $pCO_2$ from Takahashi et al. (2009) provided the target. The dataset was divided into 50% for

180FFNN training and 25% for its evaluation. This 25% did not participate in the training. This set is used to

181monitor the performance of the training process and to drive its convergence. The remaining 25% (each 4[th]

182point) of the dataset were used after training for the FFNN model validation. More details about the FFNN

183training process can be found in Rumelhart et al. (1986) and Bishop (1995). Validation and evaluation

184datasets were chosen quasi-regularly in space and time to take into account all regions and seasonal

185variability. In order to improve the accuracy of the reconstruction, the model was applied separately for

186each month. We have developed a FFNN model with 5 layers (3 hidden layers). 12 models with a common

187architecture were trained. Tests with one model for 12 months showed a slight decrease in accuracy (not

188presented here). About 17500 data were available for each month to train the model, resulting in monthly

189FFNN models with about 1856 parameters.

191(2) Step 2: reconstruction of anomalies

192During the second step, normalized $pCO_2$ anomalies were reconstructed as a nonlinear function of

193normalized SSS, SST, SSH, Chl, MLD, $xCO_2$ and their anomalies, as well as geographic position:

$$pCO_{2,anom,n} = \left( SSS_n, SST_n, SSH_n, Chl_n, MLD_n, xCO_{2,n}, \right.$$

$$\left. SSS_{anom,n}, SST_{anom,n}, SSH_{anom,n}, Chl_{anom,n}, MLD_{anom,n}, xCO_{2,anom,n}, lon_{n,1}, lon_{n,2}, lat_n \right) \quad (3)$$

195Surface ocean $pCO_2$ anomalies computed as the differences between collocated $pCO_2$ values based on

196SOCAT observations and monthly $pCO_2$ climatologies reconstructed during the first step provided the

197targets:

$$pCO_{2,anom} = pCO_{2,SOCAT} - pCO_{2,clim,FFNN} \quad (4)$$

199The set of target data was again divided into 50% for the training algorithm, 25% for evaluation and 25%

200for model validation. As in step (1) the model was trained separately for each climatological month. There

201were thus 12 models sharing a common architecture but trained on different data. At this step, in order to

202increase the amount of data during training and to introduce information on the seasonal cycle, the model

203was trained using as a target $pCO_2$ data from the month in question as well as those from the previous and

204following month during the entire period 2001-2016. Figures 1 (b) and 1 (c) show an example of data

205distribution for the sole months of January over the period 2001-2016 (Fig. 1 (b)) and for the three months

206time-window December-January-February 2001-2016 used in the training algorithm of the January FFNN

207model (Fig. 1 (c)). In this particular example, the choice of three months provided a better cover of the

208region and doubled the number of data at high latitudes.

K-fold cross-validation was used for the evaluation and the validation of the FFNN architecture. Cross-
validation relied on K=4 different subsampling of the dataset to draw 25% of independent data for
validation (Fig. S1). Each sampling fold was tested on 5 runs of the FFNN for each month. Each of these 5
runs is characterized by different initial values that are chosen randomly. From these 5 results, the best was
chosen based on root-mean-square-error (RMSE), $r^2$ and bias.

The final model architecture at step 2 had 3 layers (1 hidden layer). About 10000 samples were available
for training for each month, thus, a model with 541 parameters was developed. Note that a higher number
of parameters did not show a significant improvement of accuracy.

b) Reconstruction of surface ocean $pCO_2$

The previous section presented the development of the "optimal" architecture of a FFNN model for the
reconstruction of global surface ocean $pCO_2$, and the estimation of its accuracy. This FFNN model was
used to provide the final product for scientific analysis and comparison with other mapping approaches. In
order to provide the final output, the selected FFNN architecture is trained on all available data: 100% of
data for training, 100% for evaluation and 100% for validation. The network was executed 5 times
(different initial values) and the best model was selected based on validation results considering root-mean-
square-error (RMSE), $r^2$ and bias computed between network output and SOCAT derived surface ocean
$pCO_2$ data. The final model output is referred to as the LSCE-FFNN product.

2.3. Computation of sea-air $CO_2$ fluxes.

Sea-air $CO_2$ flux $f$ was calculated following Rödenbeck et al. (2015) as:

$$f = k\rho L \left( pCO_2 - pCO_2^{atm} \right) \qquad (5)$$

where k is the piston velocity estimated according to Wanninkhof (1992):

$$k = \Gamma u^2 \left( Sc^{CO_2} / Sc^{Ref} \right)^{-0.5} . \qquad (6)$$

The global scaling factor $\Gamma$ was chosen as in Rödenbeck et al. (2014) with the global mean $CO_2$ piston
velocity equaling to 16.5 cm/h. $Sc$ corresponds to the Schmidt number estimated according to Wanninkhof
(1992). The wind speed was computed from 6-hourly NCEP wind speed (Kalnay et al., 1996). $\rho$ is
seawater density in (5) and L is the temperature-dependent solubility (Weiss, 1974). $pCO_2$ corresponds to

the surface ocean $pCO_2$ output of the mapping method. $pCO_2^{atm}$ was derived from the atmospheric $CO_2$
mixing ratio fields provided by the Jena inversion s76_v4.1 (http://www.bgc-jena.mpg.de/CarboScope/).

243**3. Results.**

2453.1. Validation.

246The subset of data used for network validation, that is 25% of the total, represents independent observations
247as they did not participate in training during model development (see 2.2a). The skill of the FFNN to
248reconstruct monthly climatologies of surface ocean $pCO_2$, was assessed by comparing collocated
249reconstructed $pCO_2$ and corresponding values from Takahashi et al. (2009). The global climatology was
250reconstructed with a satisfying accuracy during step 1 with a RMSE of 0.17 μatm and $r^2$ of 0.93. Model
251output of step 2 was assessed by K-fold cross-validation as presented before: K=4 different subsets of
252independent data were drawn from the dataset and the network was run 5 times on each subset. From these
25320 results the best one was chosen based on RMSE, $r^2$ and mean absolute error (MAE) (the bias is
254presented in Table S1). The combination of the four best model output was used for the statistical analysis
255summarized in Table 1. Metrics were computed over the full period (2001-2016) and with reference to
256SOCAT observations (independent data only). At the global scale, the analysis yielded a RMSE of ~17.97
257μatm, while the MAE was 11.52 μatm and $r^2$ was 0.76. These results are comparable to those obtained by
258Landschützer et al. (2013) for the assessment of a surface ocean $pCO_2$ reconstruction based on an
259alternative neural network-based approach. The RMSE between SOCAT data and the climatology of $pCO_2$
260from Takahashi et al. (2009) equals 41.87 μatm, larger than errors computed for the regional comparison
261between FFNN and SOCAT (Table 1). We also estimated the RMSE for the case of 100% data used for
262training. It equals 14.8 μatm and confirms the absence of overfitting.

264Figure 2 (a) shows the time mean difference between the estimated $pCO_2$ and $pCO_2$ from SOCAT v5 data

265used for validation $mean_t \left( pCO_{2,i,j,FFNN} - pCO_{2,i,j,SOCAT} \right)$ . Large differences occurred at high
266latitudes, in equatorial regions, along the Gulf Stream and Kuroshio currents – the regions with strong
267horizontal gradients of $pCO_2$. Moreover, the standard deviation of residuals (Figure 2 (b)) in these regions
268was larger indicating that the model fails to accurately reproduce the temporal variability. The reduced skill
269of the model in these regions reflects the poor data coverage along with a strong seasonal variability (e.g.
270Southern Ocean) and/or high kinetic energy (e.g. Southern Ocean, Kuroshio and Gulf Stream currents)
271(Fig. 1 (a)). At the scale of ocean regions, (Table 1) the largest RMSE and MAE were computed for the
272Pacific Subpolar ocean (RMSE = 34.77 μatm, MAE = 23.12 μatm), while the lowest correlation coefficient
273was obtained for the equatorial Atlantic Ocean ($r^2$ = 0.57). These low scores directly reflect low data
274density and are to be contrasted with those obtained over regions with better data coverage (e.g. Subtropical
275North Pacific: RMSE = 15.86 μatm, MAE = 9.9 μatm, $r^2$ = 0.77 or Subpolar Atlantic: RMSE = 22.99 μatm,
276MAE = 15.04 μatm, $r^2$ = 0.76). Despite large time mean differences computed over the eastern Equatorial
277Pacific, scores are satisfying at the regional scale indicating error compensation by improved scores over

278the western basin (RMSE = 15.73 µatm, MAE = 10.33 µatm, $r^2$ = 0.79). Scores are low in the Southern

279Hemisphere (Table 1) and time mean differences are large (Fig. 2 (a)) reflecting sparse data coverage  (Fig.

2801 (a)).

2823.2. Qualification.

283This section presents the assessment of the final time series of reconstructed surface ocean $pCO_2$. The time

284series was computed using the best monthly models as described in section 2.2, as well as 100% of data for

285learning, evaluation and validation.

286Results of the LSCE-FFNN mapping model were compared to three published mapping methods which

287participated in the "Surface Ocean pCO2 Mapping Intercomparison" (SOCOM) exercise presented in

288Rödenbeck et al. (2015) (http://www.bgc-jena.mpg.de/SOCOM/). These methods are: (1) Jena-MLS

289oc_v1.5 (Rödenbeck et al., 2014), a statistical interpolation scheme (data-driven mixed-layer scheme;

290principal drivers used in parametrisation: ocean-internal carbon sources/sinks, SST, wind speed, mixed-

291layer depth climatology, alkalinity climatology); (2) JMA-MLR (updated version up to 2016) (Iida et al.,

2922015), based on multi-linear regressions with SST, SSS and Chl $a$ as independent variables, and (3) ETH-

293SOMFFN v2016 (Landschützer et al., 2014), a two-step neural network model with SST, SSS, MLD, Chl $a$,

294$x$CO$_2$ as drivers. The time series of $pCO_2$ and sea-air $CO_2$ flux ($f$) were assessed over 17 biomes defined by

295Fay and McKinley (2014) (Fig. 3, Table 2). These biomes were derived based on coherence in SST, Chl $a$,

296ice fraction, maximum MLD and represent regions of coherent biogeochemical dynamics.

298We followed the protocol and diagnostics proposed in Rödenbeck et al. (2015) for the comparison of the

299mapping methods between each other, respectively to observations. The following diagnostics were

300computed: (1) the relative interannual variability (IAV) mismatch $R^{iav}$ (in %) and (2) the amplitude of

301interannual variations. The relative interannual variability (IAV) mismatch $R^{iav}$ (in %) is the ratio of the

302mismatch amplitude $M^{iav}$ of the difference between the model output and observations (its temporal

303standard deviation) and the mismatch amplitude $M^{iav}_{benchmark}$ of the "benchmark". The latter was derived

304from the mean seasonal cycle of the corresponding model output where the trend of increasing yearly

305atmospheric $pCO_2$ was added (see details in Rödenbeck et al., 2015). It corresponds to a climatology

306corrected for increasing atmospheric $CO_2$, but without interannual variability.

$$R^{iav} = \frac{M^{iav}}{M^{iav}_{benchmark}} * 100\%$$

, (6)

308where

$$M^{iav} = std\left(mean\left(pCO_{2,Model} - pCO_{2,SOCAT}\right)\right) \,,$$

$$M^{iav}_{benchmark} = std\left(mean\left(D_{season}\right)\right) \,,$$

311where "mean" is a mean over the region and year and

$$312 \quad D_{season} = \left( pCO_{2,SS} + trend \left( CO_{2,atm} \right) \right) - pCO_{2,SOCAT} \quad ,$$

313$pCO_{2,SS}$ is the seasonal cycle of $pCO_2$ from the corresponding mapping method. $CO_{2,atm}$ estimates from
314$x$CO$_2$ Jena CO$_2$ inversion s76_v4.1 were used.

315$R^{iav}$ provides information on the capability of each method to reproduce the IAV compared to observations:
316a smaller $R^{iav}$ stands for better fit compared to the reference.  The amplitude of the interannual variations
317($A^{iav}$) of sea-air flux of CO$_2$ (its 2-month running mean) is estimated as the temporal standard deviation over
318the period.

3203.2.1. Interannual variability.

322The time series of globally averaged surface ocean $pCO_2$ over the period 2001-2016 are presented in Figure
3234 for LSCE-FFNN and the three other models. Surface ocean $pCO_2$ (μatm) varied between the 4 mapping
324methods in the range of ±7 μatm (Fig. 4 (a)). Modeled $pCO_2$ values were at the lower end for ETH-
325SOMFFN and JMA-MLR, while LSCE-FFNN and Jena-MLS13 computed higher values. The same
326behavior was found for 12-month running mean time series (Fig. 4 (b)). Figure 4 (c) shows the 12-month
327running mean of the difference between computed $pCO_2$ and SOCAT data (model – SOCAT) over the
328globe. JMA-MLR mostly underestimated observed $pCO_2$ with a strong interannual variability of the misfit,
329especially at the end of the period with up to -5 μatm. The difference between ETH-SOMFFN output and
330SOCAT data fluctuated in the range of ±1 μatm, with an increase in amplitude up to -2 μatm from 2010
331onward. Jena-MLS13 overestimated observations with the difference in the range of 0-1 μatm. The
332difference between LSCE-FFNN and SOCAT varies around zero between -0.7 and 1 μatm.

334The model was assessed next at biome scale. Results for all biomes are presented in the supplementary
335material (Fig. S2, S3, S4). Two biomes with contrasting dynamics are discussed hereafter in greater detail:
336(1) the Equatorial East Pacific (biome 6) characterized by a strong IAV of surface ocean $pCO_2$ and sea-air
337CO$_2$ fluxes in response to ENSO, the El Niño Southern Oscillation (Feely et al., 1999; Rödenbeck et al.,
3382015), and (2) the North Atlantic Permanently Stratified biome (biome 11) with a well-marked seasonal
339cycle, but little IAV (Schuster et al., 2013). Results for these biomes are presented in Figure 5.

341Biome 6 is relatively well-covered by observations and represents a key region for testing the skill of the
342model to reproduce the observed strong IAV linked to ENSO. El Niño events are characterized by positive
343SST anomalies, reduced upwelling and decreased surface ocean $pCO_2$ values. These episodes could be
344identified in all model time series (Fig. 5 (a)) with reduced $pCO_2$ levels in 2004/2005 and 2006/2007 (weak
345El Niño), 2002/2003 and 2009/2010 (moderate El Niño), and 2015/2016 (strong El Niño). JMA-MLR (blue
346curve) tended to underestimate $pCO_2$ during weak El Niño events. It was underestimated during the La

347Niña 2011-2012 event by Jena-MLS13. LSCE-FFNN and ETH-SOMFFN, both based on a neural network

348approach yielded similar results despite differences in network architecture and predictor datasets.

350Data coverage is particularly high over Biome 11 (Fig. 5 (b), (d), (f)). The seasonal cycle in this biome is

351dominantly driven by temperature. Modeled seasonal variability showed a good agreement across the

352ensemble of methods (Fig. 5(b)) with an increase in spring-summer and a decrease in autumn-winter.

353However, the amplitude can be different by up to 10 µatm between different models. The seasonal

354amplitude of $pCO_2$ computed by JMA-MLR increased from smaller values at the beginning of the time

355series to higher ones in the middle of the period 2005-2012. The variability of seasonal amplitude was the

356highest for Jena-MLS13 in line with the 12-month running mean time series (Fig. 5 (d)). Again, similar

357seasonal amplitude and year-to-year variability of surface ocean $pCO_2$ were obtained with LSCE-FFNN

358and ETH-SOMFFN (Fig. 5 (b), (d)). The yearly $pCO_2$ mismatch (Fig. 5 (f)) shows that observed surface

359ocean $pCO_2$ was underestimated by JMA-MLR at the beginning and at the end of the period by up to -6

360µatm, and overestimated during 2007-2011 by up to 8 µatm. Jena-MLS13 shows mostly positive

361differences in the range 0-2 µatm over the full period. LSCE-FFNN and ETH-SOMFFN vary around zero

362and between -2 – 2 µatm, being close to each other.

3643.2.2. Sea-air $CO_2$ flux variability.

366Sea-air exchange of $CO_2$ was estimated using the same gas exchange formulation (4) and wind data speed

367(6-hourly NCEP wind speed) for each mapping data (Rödenbeck et al., 2005). It is worth noting that the

368sea-air flux is sensitive to the choice of the wind speed dataset (Roobaert et al., 2018).

370Figure 6 (a) presents the global 12-month running mean of the sea-air $CO_2$ flux for four mapping methods.

371All models showed an increase in $CO_2$ uptake in response to increasing atmospheric $CO_2$ levels, albeit with

372a strong between-model variability in multi-annual trends. There is less agreement between the methods

373compared to reconstructions of surface ocean $pCO_2$ variability (Fig. 4 (b)). This results from the

374contribution of uncertainties in sea-air $CO_2$ flux estimations over regions with poor data-coverage (mostly

375in the South Hemisphere: South Pacific, South Atlantic, Indian Ocean, South Ocean; see Fig. S5).

376Nevertheless, the relative IAV mismatch was less than 30% for all methods (Fig. 6 (b)), suggesting a

377reasonable fit to observational data. The relative IAV mismatch is, however, a global score and it is biased

378towards regions with good data coverage (Rödenbeck et al., 2015). The time series reconstructed in this

379study is too short to capture decadal variations and in particular the strengthening of the sink from 2000

380onward (Landschützer et al., 2016). LSCE-FFNN computed a slowdown of ocean $CO_2$ uptake between

3812010 and 2013 with a flux of ~-1.8 GtC yr$^{-1}$ compared to ~-2.2 GtC yr$^{-1}$ for ETH-SOMFFN. A leveling-off

382was also found for JMA-MLR, albeit shifted in time. In general, the amplitudes of reconstructed $CO_2$ fluxes

383across all four methods agreed within 0.2-0.36 PgC/yr. The weighted mean of IAV (horizontal line in Fig. 6

384(b)) computed from the four methods included here was 0.25 PgC/yr. This value is close to the one of

385Rödenbeck et al. (2015) for the complete ensemble of SOCOM models (0.31 PgC/yr) estimated for the

386period 1992-2009. The largest amplitude was obtained for ETH-SOMFFN, ~0.35 PgC/yr. On the other

387hand, LSCE-FFNN has the smallest amplitude with 0.21 PgC/yr. Jena-MLS13 and JMA-MLR lie very

388close to the weighted mean value with 0.26 PgC/yr and 0.22 PgC/yr, respectively. The weighted mean and

389the dispersion of individual models around it, reflect the period of analysis (2001-2015, ETH-SOMFFN

390output provided up to 2015) and the total number of models contributing to it (see for comparison

391Rödenbeck et al., 2015). As such it does not provide information on the skill of any particular model.

393The interannual variability of reconstructed sea-air $CO_2$ fluxes (12-month running mean) showed a good

394agreement for biome 6 (East Pacific Equatorial, Fig. 7 (a)). A small discrepancy was found at the beginning

395of the period. A strong increase was computed by Jena-MLS13 for 2010-2014 that was also identified on

396$pCO_2$ variability (Fig. 5 (a)). Despite this, Jena-MLS13 has a low relative $R^{IAV}$ (26%), which confirms a

397tendency mentioned in Rödenbeck et al. (2015) that mapping products with a small relative IAV mismatch

398show larger amplitude. LSCE-FFNN and ETH-SOMFFN yielded comparable results (Fig. 7 (a), (c)) with

399relative IAV mismatches of 46% and 53%, respectively, and with amplitudes ~ 0.03 PgC/yr. Interannual

400variability reproduced by JMA-MLR falls within the range of the other models (Fig. 7 (c)), but with a $R^{IAV}$

401of ~68%.

403Reconstructed sea-air $CO_2$ fluxes over the North Atlantic Subtropical Permanently Stratified region (biome

40411) show large between model differences in amplitudes and variability. The two models based on a neural

405network show again a good agreement with $R^{IAV}$ of 17% for LSCE-FFNN and 20% for ETH-SOMFFN.

406Jena-MLS13 produced a strong seasonal variability (Fig. 7 (b)) up to 0.06 PgC/yr, and small $R^{IAV}$ of ~11%.

407Contrary to the other approaches, JMA-MLR did not reproduce a decrease in sea-air $CO_2$ at the middle of

408the period by up to 0.02 PgC/yr (Fig. 7 (b)). The model is characterized by a $R^{IAV}$ of 46% and an amplitude

409of 0.013 PgC/yr.

4113.3.3. Sea-air $CO_2$ flux trend.

413The long-term trend of sea-air $CO_2$ fluxes is dominantly driven by the increase in atmospheric $CO_2$ (see Fig.

414S7). On shorter time scales, such as for the period 2001-2016, the interannual variability at regional scales

415reflects natural modes of climate variability and local oceanographic dynamics (Heinze et al., 2015).

417Figure 8 shows the significant linear trends (p_val = 0.05) of sea-air $CO_2$ fluxes for LSCE-FFNN (a), Jena-

418MLS13 (b), ETH-SOMFFN (c) and JMA-MLR (d). A total (averaged over the globe) negative trend was

419computed for all models, albeit with large regional contrasts, and LSCE-FFNN falls within the range: Jena-
420MLS13, -0.0012 PgC/yr/yr (-0.0028 PgC/yr/yr, total value without significant t-test, Fig. S8); LSCE-
421FFNN, -0.00087 PgC/yr/yr (-0.0032 PgC/yr/yr); JMA-MLR, -0.0013 PgC/yr/yr (-0.0037 PgC/yr/yr); ETH-
422SOMFFN, -0.0025 PgC/yr/yr (-0.0059 PgC/yr/yr). LSCE-FFNN computed negative trends over most of the
423Atlantic basin, Indian Ocean and South of 40ºS, which contrasts with decreasing fluxes over the Pacific and
424locally in the Antarctic Circumpolar Current. At first order, this broad regional pattern is found in all
425models. Regional maxima and minima are, however, more pronounced in Jena-MLS13 (Fig. 8 (b)) and
426ETH-SOMFFN (Fig. 8 (c)), while a patchy distribution at sub-basin scale is diagnosed for JMA-MLR.

428The agreement in sign of computed linear trends from four models is presented in Fig. 9 (total linear trends
429without significant t-test). Over most of the ocean, all four models show very close sea-air $CO_2$ tendency. In
430the Indian Ocean (biome 14), on the other hand a positive trend was computed for JMA-MLR (0.0004
431PgC/yr/yr, and with t-test: 0.00006 PgC/yr/yr) while the three other models present a negative trend.  The
432differences between models were also found in the Pacific Ocean, especially the Southern Pacific. In the
433Eastern Equatorial Pacific region (biome 6) a total significant negative trend is presented by all models. All
434models reproduced a maximum in the southern part of biome 6 but they disagree about its amplitude and
435spatial distribution. Almost everywhere over the Atlantic Ocean the mapping methods produced the same
436sign of linear trend (Fig. 9). Only in the eastern part of the subtropical North Atlantic Jena-MLS13 gave a
437positive linear trend of $fCO_2$ (Fig. 8 (b)).

439According to LSCE-FFNN, the global ocean took up in average 1.55 PgC/yr between 2001-2015.This
440estimate is consistent with results from the other three models (Table 3) (see Table S2 for estimations per
441biomes). The spread between individual models falls in the range of the error reported in Landschützer et
442al. (2016), ±0.4-0.6 PgC/yr. Per biome, estimates of $CO_2$ sea-air fluxes provided by LSCE-FFNN are
443similarly in good agreement with those derived from the other models.

445**4. Summary and conclusion.**

447We proposed a new model for the reconstruction of monthly surface ocean $pCO_2$. The model is applied
448globally and allows a seamless reconstruction without introducing boundaries between the ocean basins or
449biomes. Our model relies on a two-step approach based on Feed-Forward Neural Networks (LSCE-FFNN).
450The first step corresponds to the reconstruction of a monthly $pCO_2$ climatology. It allows to keep the output
451of the FFNN close to the observed values in regions with poor data cover. At the second step, $pCO_2$
452anomalies are reconstructed with respect to the climatology from the first step. The model was applied over
453the period 2001-2016. Validation with independent data at global scale indicated a RMSE of 17.57 µatm, $r^2$
454of ~0.76 and an absolute bias of 11.52 µatm. In order to assess the model further, it was compared to three

455different mapping models: ETH-SOMFFN (self-organizing maps + neural network), Jena-MLS13
456(statistical interpolation), JMA-MLR (linear regression) (Rödenbeck et al., 2015). Network qualification
457followed the protocol and diagnostics proposed in Rödenbeck et al. (2015).

458Reconstructed surface ocean $pCO_2$ distributions were in good agreement with other models and
459observations. The seasonal variability was reproduced satisfyingly by LSCE-FFNN, the yearly $pCO_2$
460mismatch varied around zero, and relative IAV mismatch was 7%. LSCE-FFNN proved skillful in
461reproducing the interannual variability of surface ocean $pCO_2$ over the Eastern Equatorial Pacific in
462response to ENSO. Reductions in surface ocean $pCO_2$ during El Niño events were well reproduced. The
463comparison between reconstructed and observed $pCO_2$ values yielded a RMSE of 15.73 µatm, $r^2$ of 0.79
464and an absolute bias of 10.33 µatm over the Equatorial Pacific. The relative IAV misfit in this region was
465~17%. Despite an overall good agreement between models, important differences still exist at the regional
466scale, especially in the Southern Hemisphere and in particular, the Southern Pacific and the Indian Ocean.
467These regions suffer from poor data-coverage. Large regional uncertainties in reconstructed surface ocean
468$pCO_2$ and sea-air $CO_2$ fluxes have a strong influence on global estimates of $CO_2$ fluxes and trends.

471**Code and data availability.**

473Python code for $pCO_2$ climatology reconstruction, $1^{st}$ step of LSCE-FFNN model, python code for
474reconstruction of $pCO_2$ anomalies, $2^{nd}$ step of LSCE-FFNN model, are provided at the end of
475supplementary material.

477Time series of reconstructed surface ocean $pCO_2$ and $CO_2$ fluxes are distributed through the Copernicus
478Marine Environment Monitoring Service (CMEMS), http://marine.copernicus.eu/services-portfolio/access-
479to-products/, search keyword: MULTIOBS.

481**Author contribution.**

482ADS, MG, MV and CM contributed to the development of the methodology and designed the experiments,
483ADS carried them out. ADS developed the model code and performed the simulations. ADS prepared the
484manuscript with contributions from all co-authors.

487**Acknowledgments.**

488The authors would like to thank the two referees, Christian Rödenbeck and Luke Gregor, for their helpful
489comments and questions, as well as Frederic Chevallier and Gilles Reverdin for their suggestions. This
490study was funded by the AtlantOS project (EU Horizon 2020 research and innovation program, grant

491agreement no. 2014-633211). MV also acknowledges funding by the CoCliServ and EUPHEME projects 492(ERA4CS program).

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

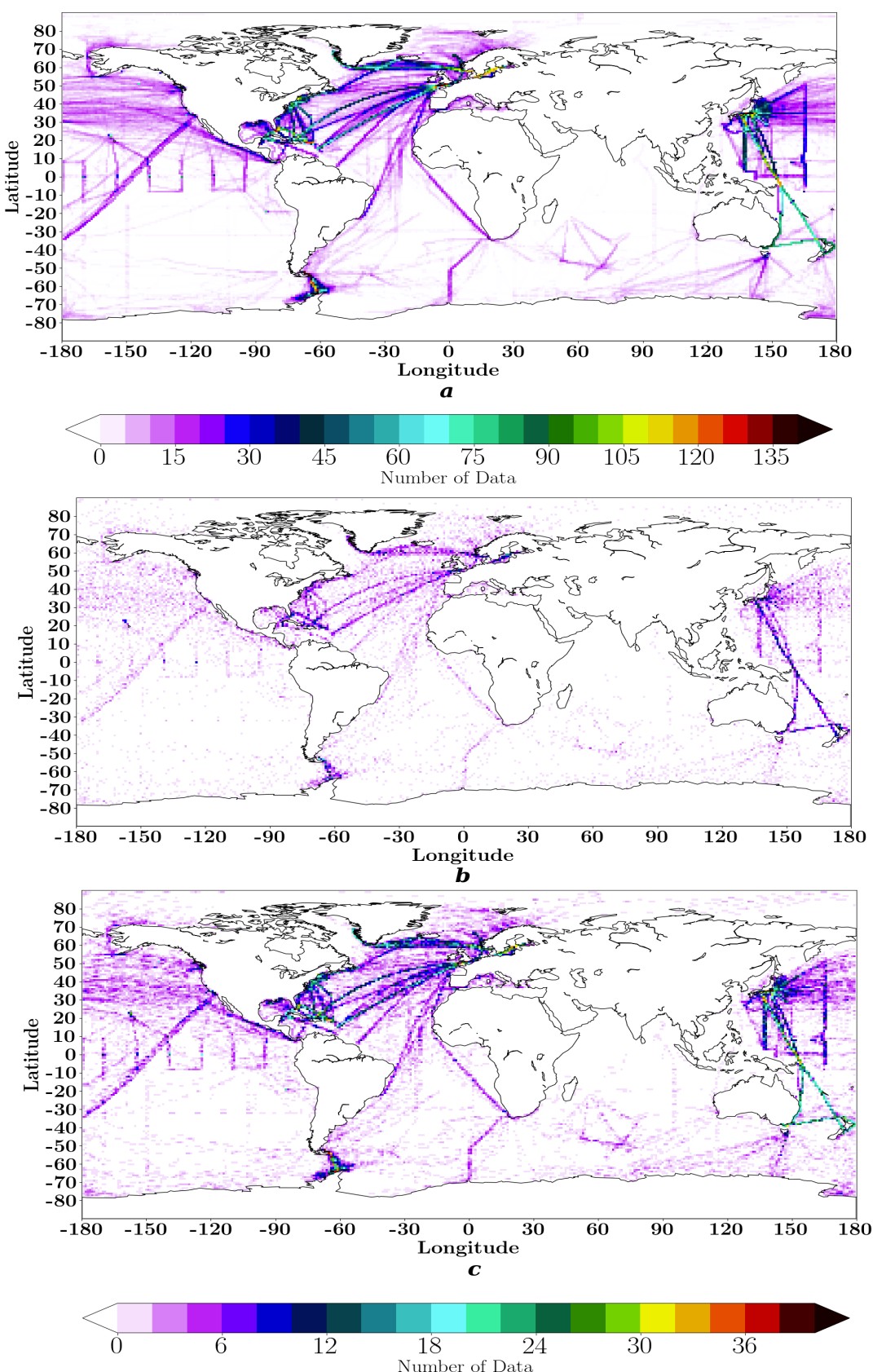

**Figure 1: Spatial distribution of SOCAT data (number of measurements per grid point): (a) - period 2001-2016; (b) - all months of January for period 2001-2016; (c) - all months of December-January-February for period 2001-2016.**

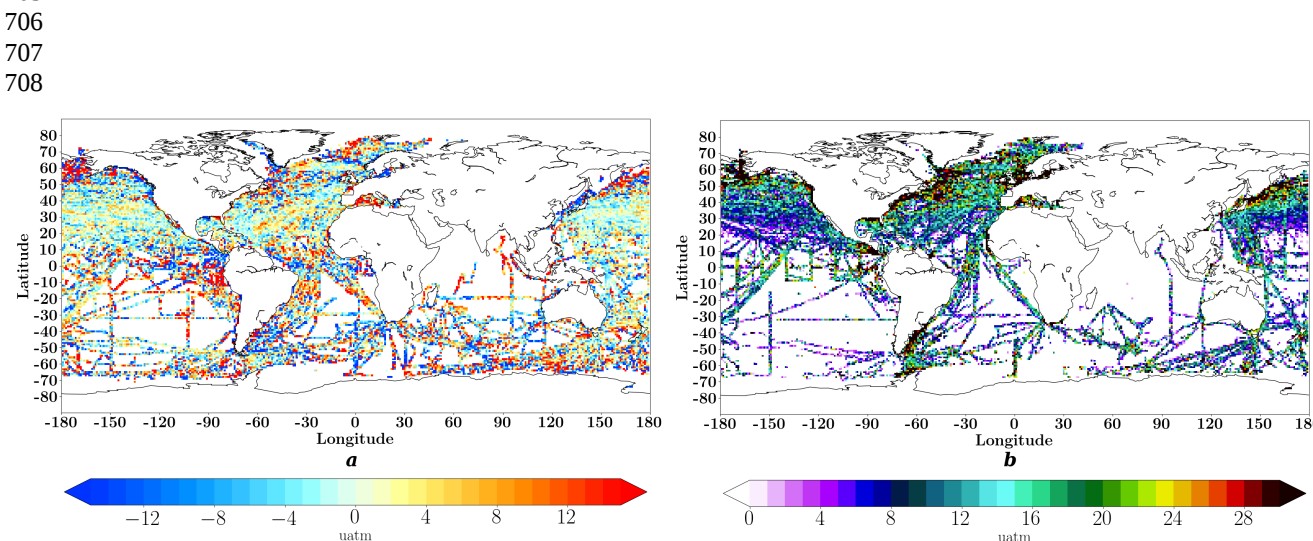

*a*

*b*

**Figure 2: Time mean differences (µatm) between monthly LSCE-FFNN pCO₂ and SOCAT pCO₂ data used for evaluation of the model over the period 2001-2016 (a) and its std (b).**

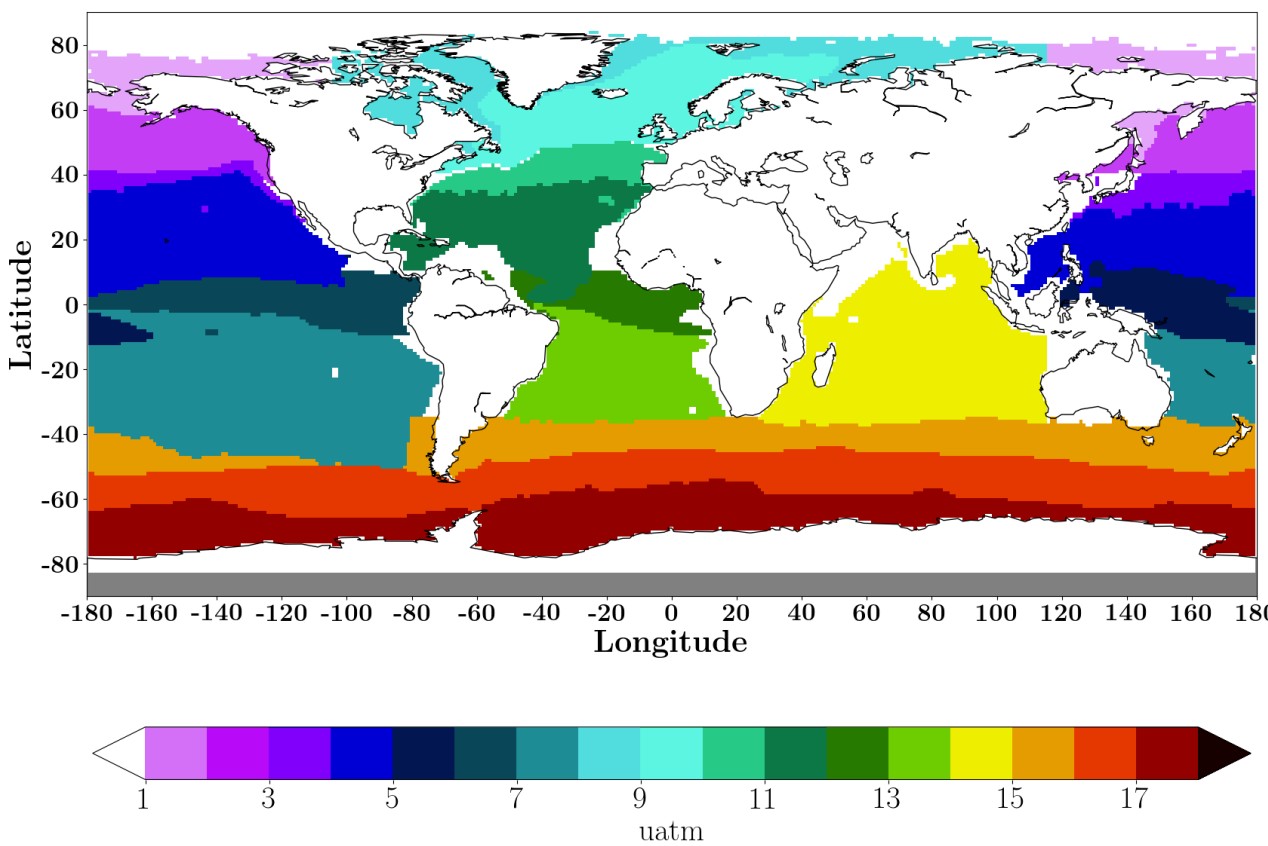

**Figure 3: Map of biomes (after Rodenbeck et al. (2015); and Fay and McKinley (2014)) used for comparison. See table 2 for biome names.**

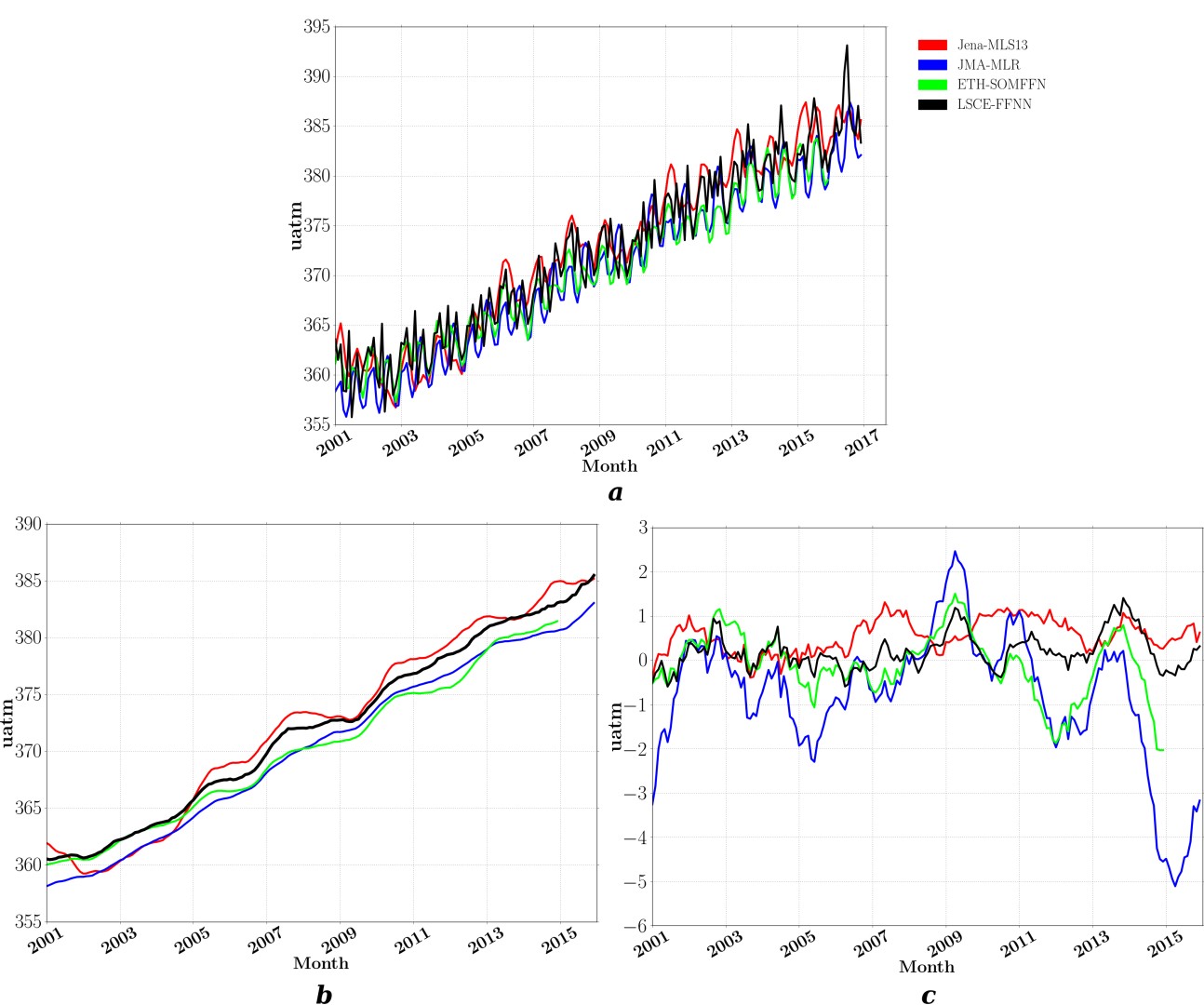

**Figure 4: Global oceanic pCO₂: black - LSCE-FFNN, blue - JMA, red - Jena, green - ETH-SOMFFN; (a) – global average monthly time series, (b) – global 12-month running mean average, (c) - yearly pCO₂ mismatch (difference of mapping methods and SOCAT data).**

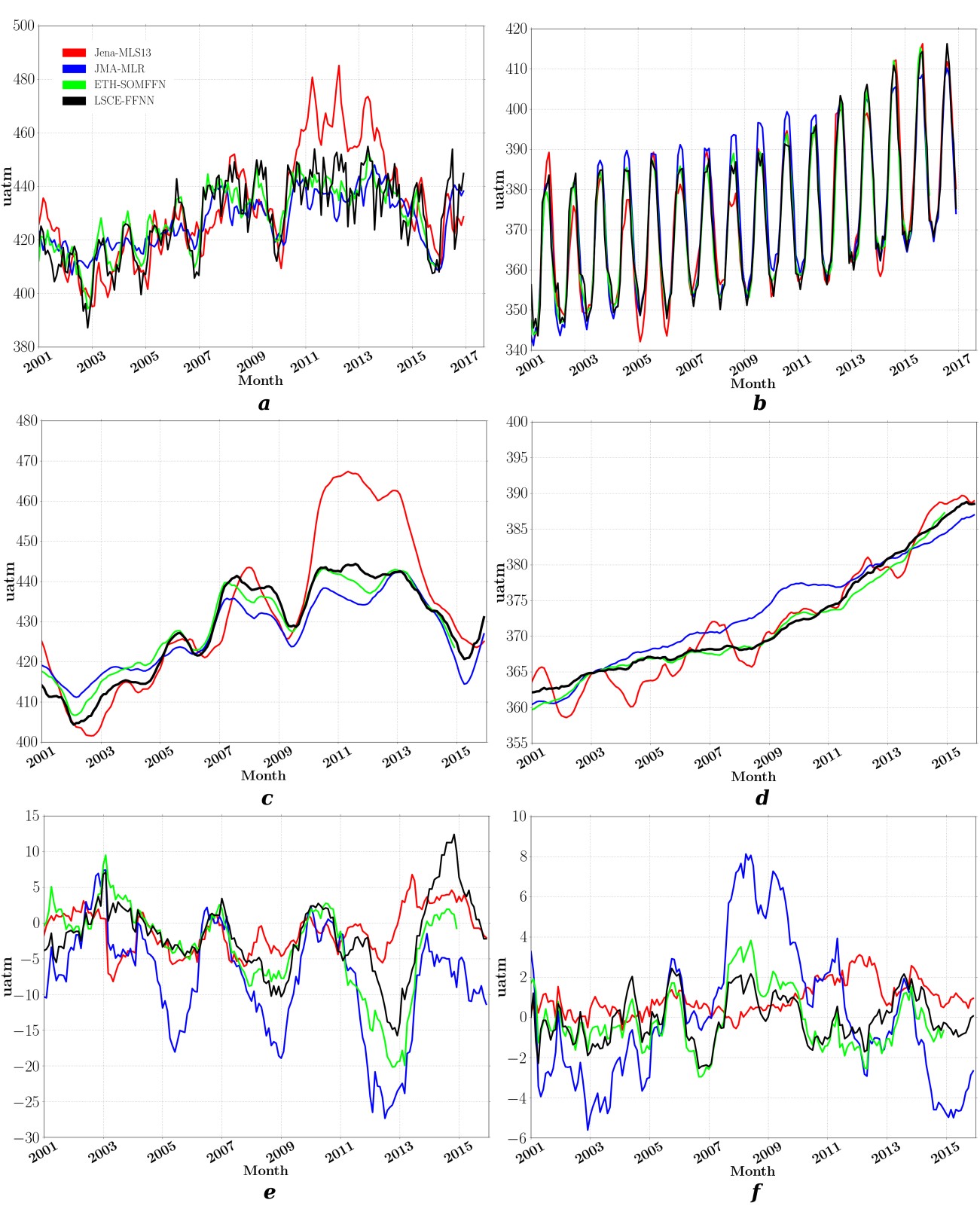

**Figure 5: Surface ocean pCO₂: Equatorial East Pacific (biome 6) (left) and Subtropical Permanently Stratified North Atlantic (biome 11) (right): black – FFNN, blue – JMA, red – Jena, green – ETH-SOMFFN; (a), (b) – monthly time series averaged over biome; (c), (d) – 12-month running mean averaged over biome; (e), (f) – yearly pCO₂ mismatch (difference of mapping methods and SOCAT data).**

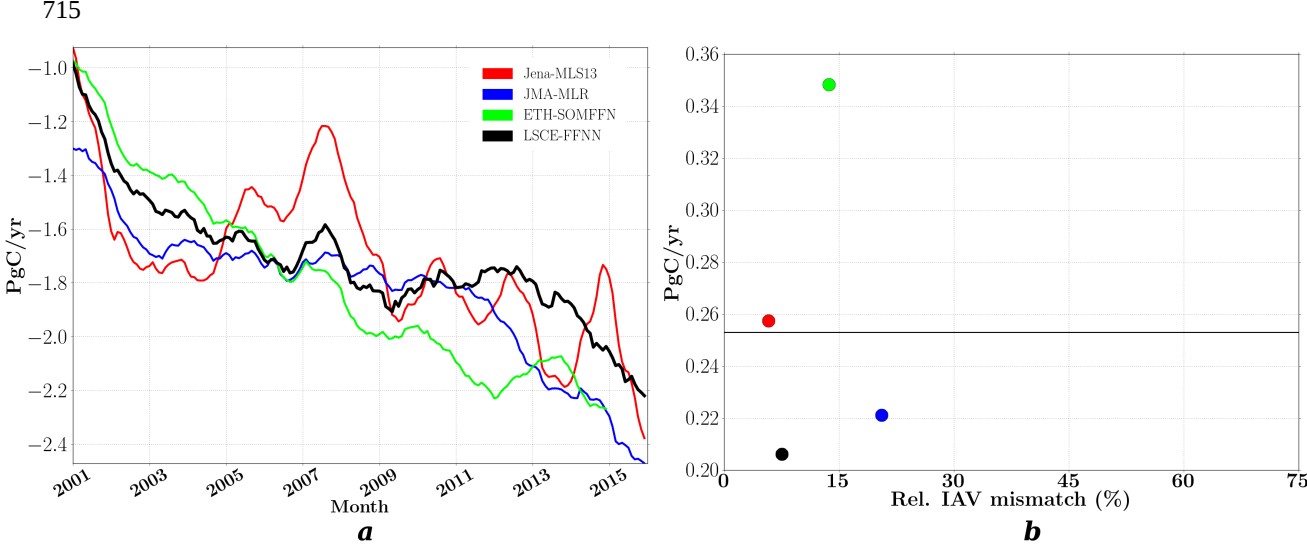

**Figure 6: (a) – Interannual global ocean sea-air CO₂ flux (12-month running mean); (b) – amplitude of interannual CO₂ flux plotted against the relative IAV mismatch amplitude. The weighted mean is given as a horizontal line.**

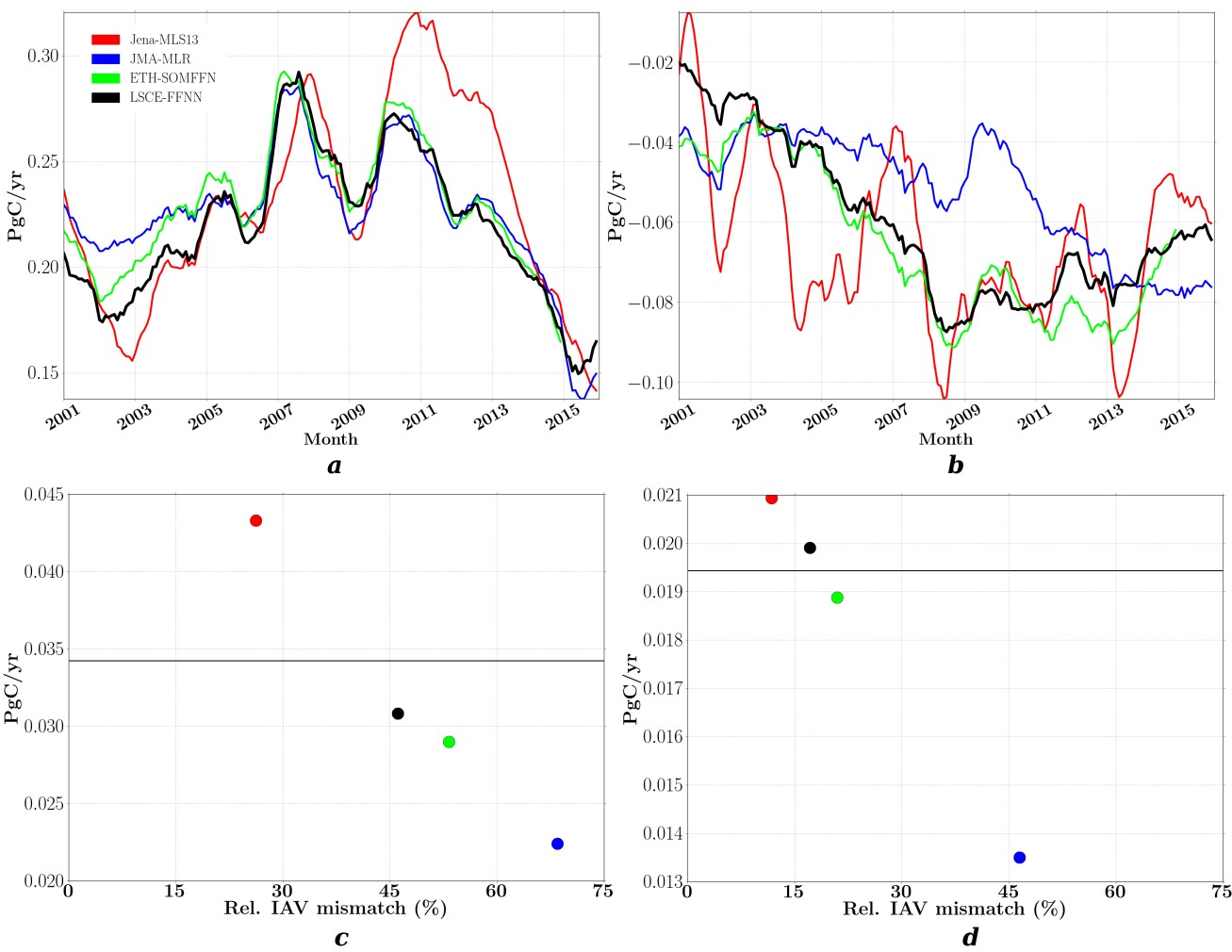

**Figure 7: Global ocean interannual sea-air CO₂ flux (12-month running mean): (a) Equatorial East Pacific (biome 6) and (b) Subtropical Permanently Stratified North Atlantic (biome 11). Amplitude of interannual CO₂ flux plotted against the relative IAV mismatch amplitude: (c) Equatorial East Pacific (biome 6) (left) and (d) Subtropical Permanently Stratified North Atlantic (biome 11). The weighted mean is given as a horizontal line.**

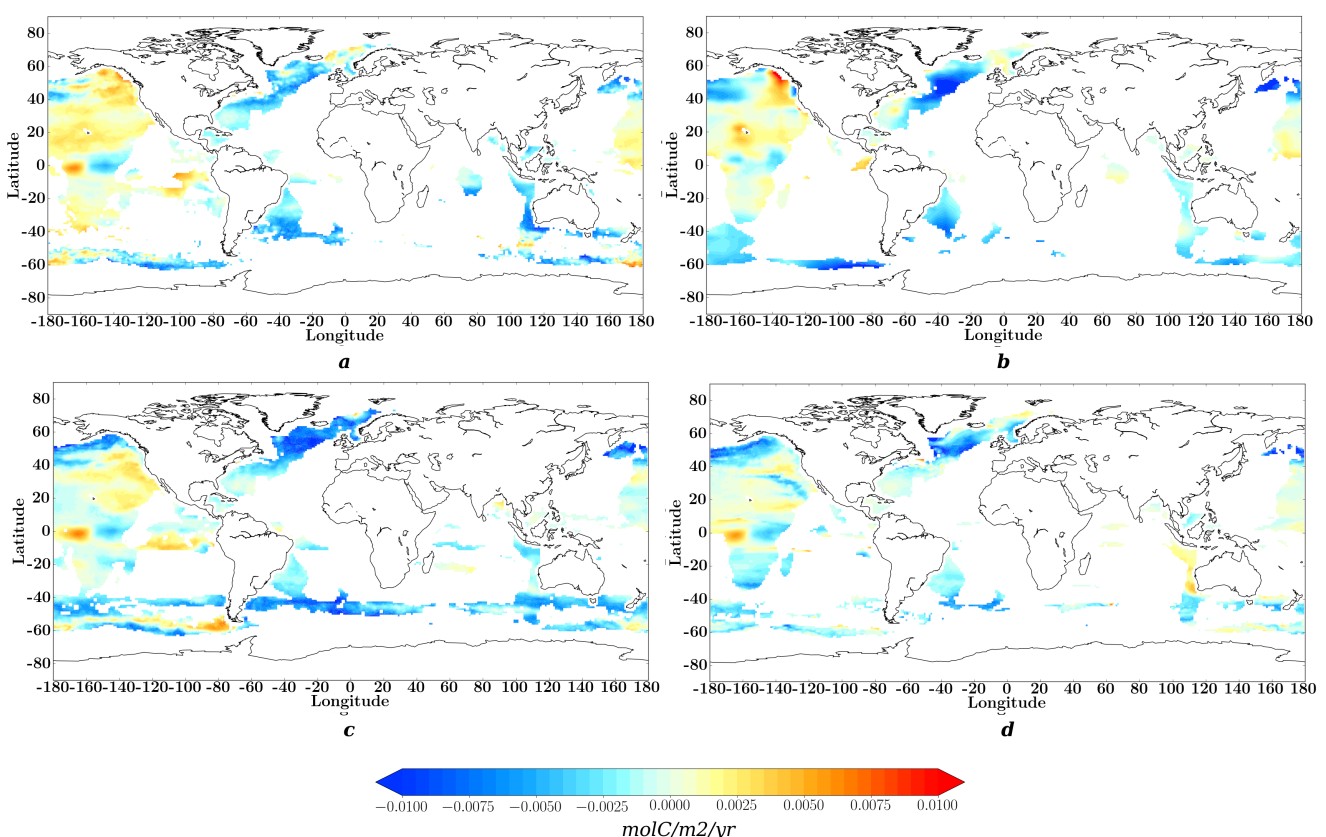

**Figure 8: Significant (p_val = 0.05) linear trend of fCO₂ for common period 2001-2015: (a) – LSCE-FFNN; (b) – Jena-MLS13; (c) – ETH-SOMFFN; (d) – JMA-MLR.**

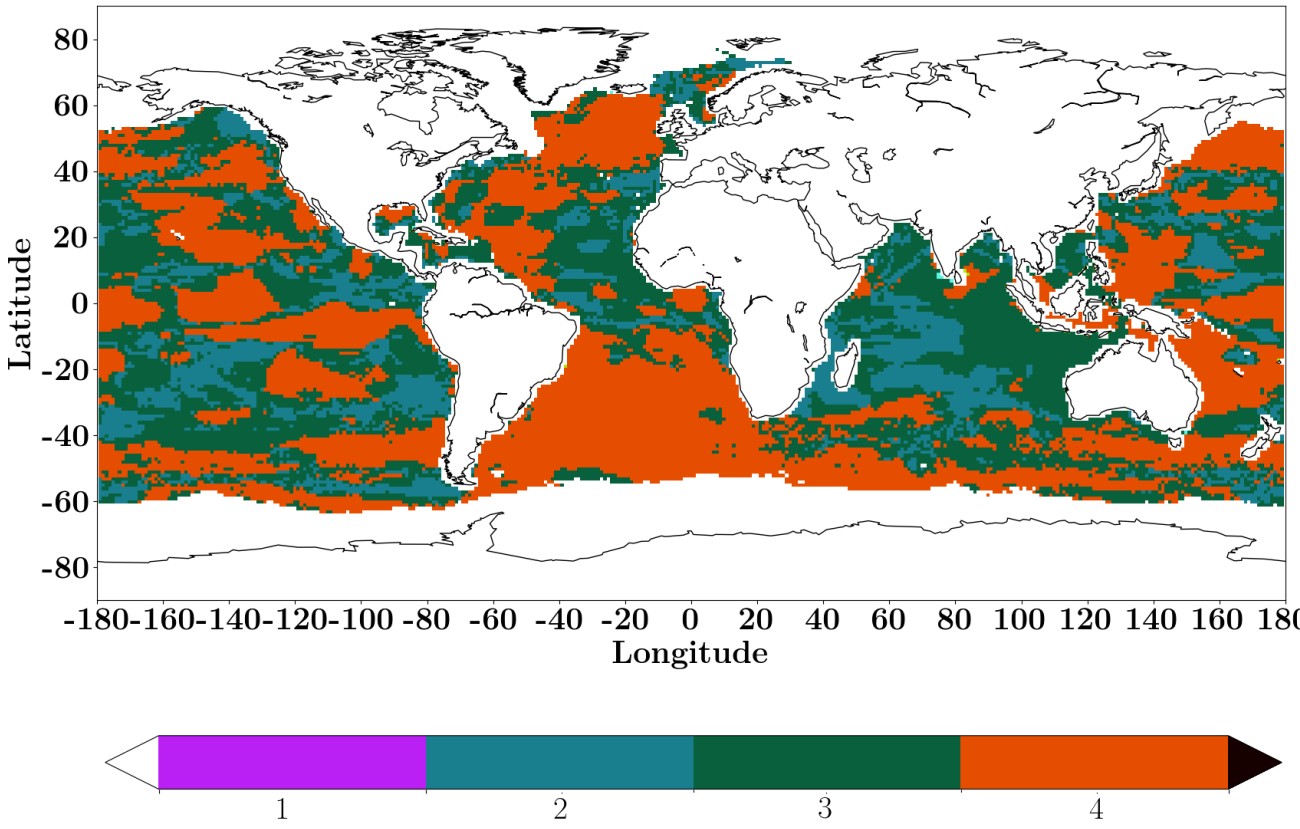

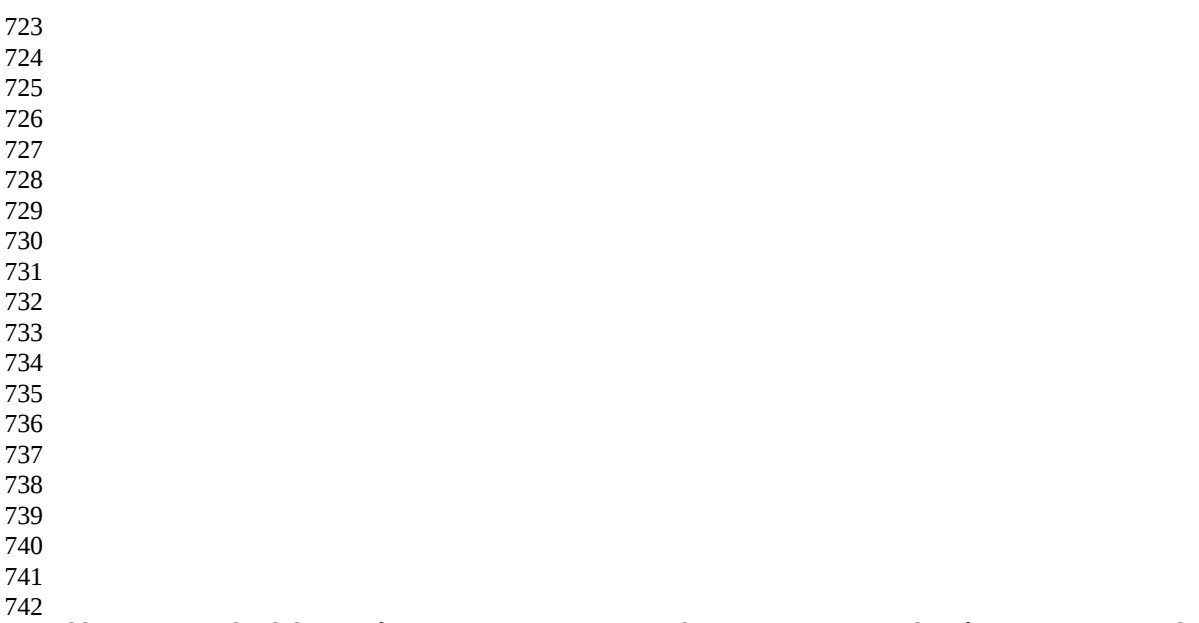

**Figure 9: Agreement between four mapping methods in their linear trend of sea-air CO₂ flux. Color-bar represents the number of products that have the same sign of linear trend.**

743Table 1: Statistical validation of LSCE-FFNN. Comparison between reconstructed surface ocean pCO₂ and

744pCO₂ values from SOCAT v5 database not used in the training algorithm for the period 2001-2016 over the

745global ocean (except for regions with ice-cover) and for large oceanographic regions. In round brackets:
746number of measurements per region

| Model | Latitude boundaries | RMSE (µatm) | $r^2$ | MAE (µatm) |
|---|---|---|---|---|
| FFNN Global | | 17.97 | 0.76 | 11.52 |
| Arctic (150) | 76ºN to 90ºN | 22.05 | 0.54 | 17.1 |
| Subpolar Atlantic (21903) | 49ºN to 76ºN | 22.99 | 0.76 | 15.04 |
| Subpolar Pacific (4529) | 49ºN to 76ºN | 34.77 | 0.65 | 23.12 |
| Subtropical Atlantic (41331) | 18ºN to 49ºN | 17.28 | 0.69 | 11.27 |
| Subtropical Pacific (41867) | 18ºN to 49ºN | 15.86 | 0.77 | 9.9 |
| Equatorial Atlantic (7300) | 18ºS to 18ºN | 17.27 | 0.57 | 11.44 |
| Equatorial Pacific (27092) | 18ºS to 18ºN | 15.73 | 0.79 | 10.33 |
| South Atlantic (3002) | 44ºS to 18ºS | 17.81 | 0.63 | 12.28 |
| South Pacific (12934) | 44ºS to 18ºS | 13.52 | 0.63 | 9.36 |
| Indian Ocean (2871) | 44S to 30N | 17.25 | 0.62 | 11.6 |
| Southern Ocean (16334) | 90ºS to 44ºS | 17.4 | 0.58 | 11.92 |

748Table 2: Biomes from Fay and McKinley (2014) used for time series comparison (Fig. 3)

| Number | Name |
|---|---|
| 1 | (Omitted) North Pacific Ice |
| 2 | Subpolar Seasonally Stratified North Pacific |
| 3 | Subtropical Seasonally Stratified North Pacific |
| 4 | Subtropical Permanently Stratified North Pacific |
| 5 | Equatorial West Pacific |
| 6 | Equatorial East Pacific |
| 7 | Subtropical Permanently Stratified South Pacific |
| 8 | (Omitted) North Atlantic Ice |
| 9 | Subpolar Seasonally Stratified North Atlantic |
| 10 | Subtropical Seasonally Stratified North Atlantic |
| 11 | Subtropical Permanently Stratified North Atlantic |
| 12 | Equatorial Atlantic |
| 13 | Subtropical Permanently Stratified South Atlantic |
| 14 | Subtropical Permanently Stratified Indian Ocean |

| 15 | Subtropical Seasonally Stratified Southern Ocean |
| 16 | Subpolar Seasonally Stratified Southern Ocean |
| 17 | Southern Ocean Ice |

Table 3: Mean of sea-air $CO_2$ flux (PgC/yr) over the Global Ocean and per regions for period in common
(2001-2015). Averages over the period 2001-2009 are presented between brackets. The last column
presents a comparison to best estimates from Schuster et al. (2013) for the Atlantic Ocean (1990 – 2009).

| Region | Latitude boundaries | LSCE-FFNN | ETH-SOMFFN | Jena-MLS13 | JMA-MLR | Schuster et al. (2013), 1990-2009 |
|---|---|---|---|---|---|---|
| Global | | -1.55 (-1.44) | -1.67 (-1.47) | -1.55 (-1.41) | -1.74 (-1.62) | --- |
| Arctic | 76ºN to 90ºN | -0.001 | -0.001 | -0.001 | -0.001 | -0.12±0.06 |
| Subpolar Atlantic | 49ºN to 76ºN | -0.15 (-0.15) | -0.14 (-0.12) | -0.15 (-0.15) | -0.16 (-0.15) | -0.21±0.06 |
| Subpolar Pacific | 49ºN to 76ºN | -0.003 (-0.005) | -0.009 (-0.004) | -0.006 (-0.004) | -0.027 (-0.021) | --- |
| Subtropical Atlantic | 18ºN to 49ºN | -0.21 (-0.19) | -0.21 (-0.19) | -0.2 (-0.18) | -0.21 (-0.2) | -0.26±0.06 |
| Subtropical Pacific | 18ºN to 49ºN | -0.45 (-0.46) | -0.49 (-0.48) | -0.47 (-0.46) | -0.49 (-0.47) | --- |
| Equatorial Atlantic | 18ºS to 18ºN | 0.085 (0.09) | 0.085 (0.095) | 0.08 (0.082) | 0.1 (0.11) | 0.12±0.04 |
| Equatorial Pacific | 18ºS to 18ºN | 0.42 (0.41) | 0.4 (0.4) | 0.44 (0.42) | 0.38 (0.37) | --- |
| South Atlantic | 44ºS to 18ºS | -0.17 (-0.16) | -0.18 (-0.16) | -0.18 (-0.17) | -0.23 (-0.22) | -0.14±0.04 |
| South Pacific | 44ºS to 18ºS | -0.33 (-0.34) | -0.4 (-0.39) | -0.35 (-0.34) | -0.49 (-0.47) | --- |
| Indian Ocean | 44S to 30N | -0.25 (-0.2) | -0.32 (-0.29) | -0.27 (-0.26) | -0.27 (-0.29) | --- |
| Southern Ocean | 90ºS to 44ºS | -0.38 | -0.29 | -0.36 | -0.26 | --- |
