# Peer review of "1LSCE-FFNN-v1: A two-step neural network model for the 2reconstruction of surface ocean pCO2 over the Global 3Ocean."

_Geoscientific Model Development, 2018_

## Short Comment (SC1) · 7 Nov 2018

Dear authors,

in my role as Executive editor of GMD, I would like to bring to your attention our Editorial version 1.1:

http://www.geosci-model-dev.net/8/3487/2015/gmd-8-3487-2015.html

This highlights some requirements of papers published in GMD, which is also available on the GMD website in the 'Manuscript Types' section:

http://www.geoscientific-model-development.net/submission/manuscript_types.html

[Figure]

In particular, please note that for your paper, the following requirement has not been met in the Discussions paper:

- "The main paper must give the model name and version number (or other unique identifier) in the title."

Please add the version number of FFNN-LSCE to the title of your manuscript upon revision.

As explained in https://www.geoscientific-model-development.net/about/manuscript_types.html GMD is encouraging that authors upload the program code of models (including relevant data sets) as a supplement or make the code and data or the exact model version described in the paper accessible through a DOI (digital object identifier). In case your institution does not provide the possibility to make electronic data accessible through a DOI you may consider other providers (eg. zenodo.org of CERN) to create a DOI. Please note that in the code accessibility section you can still point the reader how to obtain the newest version.

Best, Astrid Kerkweg

---

## Referee Comment (RC1) · Rödenbeck (Referee) · 30 Nov 2018

The authors present a method to interpolate temporally and spatially discrete surface-ocean pCO2 observations into a gridded field, using non-linear regression against various environmental drivers. Compared to similar methods, two specific features are (a) estimating seasonality separately based on data also outside the calculation period, thereby improving its data constraint, and (b) the addition of SSH to the set of predictors. This is an interesting contribution to the ensemble of existing interpolation methods, enlarging the range of plausible outcomes. The manuscript nicely describes the details of the method, and presents evaluation metrics, also in the context of other

methods from the SOCOM ensemble. I clearly recommend to publish this manuscript. Below some suggestions mainly to further improve clarity at a few places.

Comments:

The method is very similar to CARBONES-NN (not published in the peer-reviewed literature but described in the SOCOM paper Rödenbeck et al., 2015), developed at LSCE as well. I therefore assume that the presented method builds on and supersedes CARBONES-NN. Is this correct? If so, this is an interesting piece of information to users of the SOCOM ensemble and should be mentioned in this paper. Further, I would find it fair to give credit to the authors of CARBONES-NN (to my knowledge, Abdou Kane and Philippe Peylin).

On reading the manuscript, a few open questions arose that I'd find interesting to consider, either for this paper (in case you already made some corresponding tests) or in a follow-up study. (1) You mention the inclusion of SSH, but do not discuss it. How does it influence, and possibly improve, the results? (2) How much do the data outside the period improve the estimates of seasonality from the 1st step? (3) Couldn't step 1 also be fitted against the SOCAT data (folded into a climatological year as done by Takahashi et al., 2009) as well, rather than against the already-interpolated climatology? This would also bring in the data from the most recent years not yet in the Takahaski et al. (2009) climatology.

Title: While there is no question that the authors are fully free to choose a name for their method, it seems that they intended to use a "SOCOM-style" name. Therefore I'd like to point out that all names in the SOCOM paper are of the form INSTITUTION-METHOD, not the other way round.

Line 78: "On a larger data set" - please clarify what this means

Lines 79-81 and 446: Unclear statement - isn't it the 2nd step (not the 1st) which takes care of changes?

Line 114: If you took surface pressure from the s76_v4.1 run, it is in fact from NCEP (Kalnay et al.). This should be mentioned.

Lines 141, 142, 191: The "1" and "2" are awkward, and should rather be part of the subscript.

Line 174: Isn't the 1st step using climatologies of the driver variables? Please clarify. Also, is it correct that the 1st step uses un-normalized drivers (rather than SST_n etc.)?

Line 191: I assume that the "n" in the subscript to pCO2 is not correct, is it?

Line 197: Maybe say "for each climatological month".

Line 209: Does "chosen automatically" mean randomly? Please clarify.

Line 212: What does "final model" mean? Is it the model of step 2?

Line 236: "(4)" should probably be "(5)".

Line 238: Is it from run s76_v4.1 as well? Please mention.

Line 244: Isn't "did not participate" in contradiction to Sect 2.2 b) which says "100% used for training"?

Line 283: Though the Jena scheme indeed uses SST etc in its parameterizations, the sentence can be misunderstood as meaning regression drivers. Maybe just remove the list of drivers here.

Line 285: I feel "combined" is confusing and can be removed.

Line 286: Please mention whether you used the same versions of these methods as used in the SOCOM paper, or whether you used updated versions. If updated, the respective version IDs should be given.

Line 286-287: The sentence "Qualification...(2015)." is repeated in the next paragraph and may be deleted here. I feel the next sentence "The time series..." should start a new paragraph.

Line 392: I guess "small IAV" should be "small relative IAV mismatch", isn't it?

Line 394, 296, 403, 455: The figures after the decimal point are certainly meaningless and should be removed.

Lines 412, 425: Is is unclear to me what "total negative trend" means.

Line 413 and below: "PgC/yr" is the unit of a flux, not of a flux trend. Please clarify the unit (it should be something like "PgC/yr/yr" or "PgC/yr/decade").

Lines 421ff: Are trends like 0.0004PgC/yr/yr statistically significant given the IAV? If not, it is not actually appropriate to call them "positive".

Figures: In my print-out, the annotations and labels of most figures are much too small to be readable.

Fig 9, color bar: In this figure, the numbers should be within the colors, not between them, otherwise the meaning is ambigious.

Fig S7 caption: "according" probably means "agreement"?

Typos:

Line 69: "Feed Forward"

Line 135: the overbar of mean SSS should stretch over all three letters

Line 270: "bias" (2 times)

Line 287: "CO2 flux"

Line 297: "latter"

Line 391: Add comma after "Despite this"

Fig 2 caption: "(a)" seems misplaced. It could be after "2016", in accordance with placement of "(b)" after the respective desction.

[Figure]

---

## Referee Comment (RC2) · Luke Gregor (Referee) · 5 Feb 2019

The study introduces a new method that calculates pCO2 as climatology + anomaly. I think that this is a novel approach and is definitely relevant to the community. The authors do a good job of explaining the method and comparing it with past studies. I really like the approach and the method seems to perform well relative to the other methods in the SOCOM ensemble. I enjoyed reading the manuscript; however, there are many typos scattered throughout the document. I have noted most of these with track changes in the PDF document using Adobe Reader (attached as "supplement"). Below are some general comments, then repeated mistakes that can be fixed with find/replace.
* * *
General notes

The authors limit their study from 2001 to 2016. I realise that data is sparse before 2001, but it would be interesting to see how the model fares. It is useful for the community to know if this method (and in general the climatology-regression approach) can predict in data-sparse periods. It seems to be good at predicting in data-sparse regions.

It is really good that the authors use quasi-regular spacing to define the train-test-validation splits, but I would like to know more about how they defined these splitting points. On this point, I also think it would be useful to have a measure of how much a model overfits. The authors could compare the test RMSE with the RMSE of the model trained with 100% of the data. If the latter is much smaller than the 50:25:25 splits, then there is overfitting. This is important for when the method is compared with other gap-filling methods.

Regarding the figures, I struggle to see the difference between the brown and the black lines as I'm slightly colourblind. I would like to see another colour for the Jena-MLS method.
* * *
Repeated mistakes:

- absolute bias = mean absolute error (MAE = abs($\hat{y} - y$).mean()) – though I would like to see biases too ($\hat{y} - y$)

- data base = database

- data set = dataset

- southern/northern hemisphere = Southern/Norther Hemisphere

Please also note the supplement to this comment:
https://www.geosci-model-dev-discuss.net/gmd-2018-247/gmd-2018-247-RC2-supplement.pdf

―――――――――――――――

[Figure]

**Supplement:**

[revised manuscript text omitted]

(Fig. 1(a)).

The following formula is used to convert $f$CO$_2$ to pCO$_2$ (Körtzinger et al., 1999):

$$fCO_2 = pCO_2 \exp\left( p\, \frac{B+2\delta}{RT} \right) ,\ (1)$$

where $f$CO$_2$ and pCO$_2$ are in µatm, p is the total pressure (Pa), R=8.314 JK$^{-1}$ is the gas constant, T is the absolute temperature (K). Parameter B (m$^3$mol$^{-1}$) is estimated as: B = (-1636.75 + 12.0408 T – 3.27957 *

$10^{-2}$ T$^2$ + 3.16528 * $10^{-5}$ T$^3$) $10^{-6}$ . The parameter δ is the cross virial coefficient (m$^3$mol$^{-1}$): δ = (57.7 –

0.118T) $10^{-6}$ . The total pressure is from the Jena  (6h, 5ºx5º) (http://www.bgc- jena.mpg.de/CarboScope/?ID=s).

Monthly global observed physics reprocessed products distributed through the Copernicus Marine

Environment Monitoring Service (CMEMS) (0.25ºx0.25º) (http://marine.copernicus.eu/services- portfolio/access-to-products/?

option=com_csw&view=details&product_id=MULTIOBS_GLO_PHY_REP_015_002) were used for SSS,

SST and SSH. The GlobColour project provided monthly CHL distributions at 1ºx1º resolution (http://www.globcolour.info/products_description.html). For MLD, daily data from the "Estimating the

Circulation and Climate of the Ocean" (ECCO2) project Phase II, at 0.25ºx0.25º resolution (Menemenlis et al., 2008) were used. For $x$CO$_2$ atmospheric, the 6h data from Jena CO$_2$ inversion s76_v4.1 on a 5ºx5º grid selected (http://www.bgc-jena.mpg.de/CarboScope/?ID=s). Finally, an ice mask based on daily

"Operational Sea Surface Temperature and Sea Ice Analysis" (OSTIA) with a gridded 0.05ºx0.05º

resolution (Donlon et al., 2011) was applied.

MLD and CHL were log-transformed before their use in the FFNN algorithm because of their skewed distribution. In regions with no CHL data (high latitudes in winter) log(CHL) = 0 was applied. It does not introduce discontinuities since log(CHL) is close to zero in the adjacent region.

All data were averaged or interpolated on a 1ºx1º grid and, depending on the resolution of the data set, averaged over the month. It is worth noting that all data sets have to be normalized (i.e. centered to zero- mean and reduced to unit standard deviation) before their use in the FFNN algorithm, for example:

$$SSS_n = \frac{SSS - S\bar{S}S}{std(SSS)} .$$

Normalization ensures that all predictors fall within a comparable range and therefore avoids giving more weight to predictors with large variability ranges (Kallache et al., 2011).

As surface ocean pCO$_2$ also varies spatially, geographical positions (lat, lon) were included as predictors. In order to normalize (lat, lon) the following transformation is proposed:

$$lat_n = \sin(lat)$$

$lon_n 1 = \sin(lon)$

$lon_n 2 = \cos(lon)$

Two functions *sin* and *cos* for longitudes are used to preserve its periodical 0 to 360 degrees behavior and
thus to consider the difference of positions before and after the 0º longitude. For step 2, data required for
training were co-located at the SOCAT data positions that are used as target for the FFNN model. Details
are provided in the next section.

2.2. Method.

a) Network configuration and evaluation protocol

In this work we use Keras, a high-level neural network Python library ("Keras: The Python Deep Learning
library", Chollet, 2015; https://keras.io) to build and train the FFNN models. The identification of an
optimal configuration is the first step in the FFNN model building. This includes: the choice of number and
size of hidden layers (i.e., intermediate layers between input and output layers), connection type, activation
functions, loss function and optimization algorithm, as well as the learning rate and other
parameters. Based on a series of tests and their statistical results (RMS, correlation, bias) a hyperbolic
tangent was chosen as an activation function for neurons in hidden layers, and a linear function for the
output layer. As optimization algorithm the mini-batch gradient descent or RMSprop was used (adaptive
learning rates for each weight, Chollet, 2015; Hinton et al., 2012). The number of layers and neurons
depends on the problem. For totally connected layers (i.e., a neuron in a hidden layer is connected to all
neurons in the precedent layer and connects all neurons in the next one), the case here, it is enough to have
only one single hidden layer but two or more can help the approximation of complex functions (or complex
relations between the input and the output of the problem).

The number of the FFNN layers and number of neurons depends on one side on the complexity of the
problem: the more layers and neurons, the better the accuracy of output. However, the size also depends on
the number of patterns (data) used for training. There is a well-known empirical rule advising to have a
factor of 10 between number of patterns (data) and number of connections, or weights to adjust. This limits
the size, the number of parameters and incidentally the number of neurons, of the FFNN. This empirical
rule was followed in this study.

(1) Step 1: reconstruction of monthly climatologies

FFNN reconstructs a monthly surface ocean $pCO_2$ climatology as a nonlinear function of SSS, SST, SSH,
Chl, MLD and geographical position (longitude, latitude):

[Figure]

[Figure]

$$pCO_{2,n} = \left( SSS_n, SST_n, SSH_n, Chl_n, MLD_n, lon_n, lat_n \right) \quad (2)$$

Surface ocean $pCO_2$ from Takahashi et al. (2009) provided the target. The data set was divided into 50% for
FFNN training and 25% for its evaluation. This 25% did not participate in the training. This set is used to
monitor process performance and drive convergence. The remaining 25% (each 4th point) of the data set
were used after training for the FFNN model validation. More details about the FFNN training process can
be found in Rumelhart et al. (1986) and Bishop (1995). Validation and evaluation data sets were chosen
quasi-regularly in space and time to take into account all regions and seasonal variability. In order to
improve the accuracy of the reconstruction, the model was applied separately for each month. Tests with
one model for 12 months showed a slight decrease in accuracy (not presented here). We have developed a
FFNN model with 5 layers (3 hidden layers). About 17500 data were available for each month to train the
model, resulting in monthly FFNN models with about 1856 parameters.

(2) Step 2: reconstruction of anomalies

During the second step, $pCO_2$ anomalies were reconstructed as a nonlinear function of SSS, SST, SSH, Chl,
MLD, $xCO_2$ and their anomalies, as well as geographic position:

$$pCO_{2,n,anom} = \left( SSS_n, SST_n, SSH_n, Chl_n, MLD_n, xCO_{2,n}, \right.$$
$$\left. SSS_{anom,n}, SST_{anom,n}, SSH_{anom,n}, Chl_{anom,n}, MLD_{anom,n}, xCO_{2,anom,n}, lon_n 1, lon_n 2, lat_n \right) \quad (3)$$

Surface ocean $pCO_2$ anomalies computed as the differences between collocated $pCO_2$ values based on
SOCAT observations and monthly $pCO_2$ climatologies reconstructed during the first step provided the
targets:

$$pCO_{2,anom} = pCO_{2,SOCAT} - pCO_{2,clim,FFNN} \quad (4)$$

The set of target data was again divided into 50% for the training algorithm, 25% for evaluation and 25%
for model validation. As in step (1) the model was trained separately for each month. There were thus 12
models sharing a common architecture but trained on different data. At this step, in order to increase the
amount of data during training and to introduce information on the seasonal cycle, the model was trained
using as a target $pCO_2$ data from the month in question as well as those from the previous and following
month during the entire period 2001-2016. Figures 1 (b) and 1 (c) show an example of data distribution for
the sole months of January over the period 2001-2016 (Fig. 1 (b)) and for the three months time-window
December-January-February 2001-2016 used in the training algorithm of the January FFNN model (Fig. 1
(c)). In this particular example, the choice of three months provided a better cover of the region and
doubled the number of data at high latitudes.

K-fold cross-validation was used for evaluation and validation of the FFNN architecture. Cross-validation
relied on K=4 different subsampling of the data set to draw 25% of independent data for validation (Fig.
S1). Each sampling was tested on 5 runs of the FFNN for each month. Each of these 5 runs is characterized

by different initial values that are chosen automatically. From these 5 results, the best was chosen based on root-mean-square-error (RMSE), $r^2$ and bias.

The final model architecture had 3 layers (1 hidden layer). About 10000 samples were available for training for each month, thus, a model with 541 parameters was developed. Note that a higher number of parameters did not show a significant improvement of accuracy (not shown).

b) Reconstruction of surface ocean $pCO_2$

The previous section presented the  for the reconstruction of global surface ocean $pCO_2$, and the estimation of its accuracy.

. This FFNN model was used to provide the final product for scientific analysis and comparison with other mapping approaches. In order to provide the final output, the selected FFNN architecture is trained on all available data: 100% of data for training, 100% for evaluation and 100% for validation. The network was executed 5 times (different initial values) and the best model was selected based on validation results considering root-mean-square-error (RMSE), correlation and bias computed between network output and SOCAT derived surface ocean $pCO_2$ data. The final model output is referred to as the FFNN-LSCE product.

2.3. Computation of sea-air $CO_2$ fluxes.

Sea-air $CO_2$ flux $f$ was calculated following Rödenbeck et al. (2015) as:

$$f = k\rho L\left(pCO_2 - pCO_2^{atm}\right) \qquad (5)$$

where k is the piston velocity estimated according to Wanninkhof (1992):

$$k = \Gamma u^2\left(Sc^{CO_2}/Sc^{Ref}\right)^{-0.5}. \qquad (6)$$

The global scaling factor $\Gamma$ was chosen as in Rödenbeck et al. (2014) with the global mean $CO_2$ piston velocity equaling to 16.5 cm/h. $Sc$ corresponds to the Schmidt number estimated according to Wanninkhof (1992). The wind speed was computed from 6-hourly NCEP wind speed (Kalnay et al., 1996). $\rho$

seawater density in (4) and L  temperature-dependent solubility (Weiss, 1974). $pCO_2$ corresponds to the surface ocean $pCO_2$, output of the mapping method. $pCO_2^{atm}$ was derived from the atmospheric $CO_2$

mixing ratio fields provided by the Jena inversion (http://www.bgc-jena.mpg.de/CarboScope/).

**3. Results.**

3.1. Validation.

The subset of data used for network validation, that is 25% of the total, represents independent observations

244as they did not participate in training. The skill of the FFNN to reconstruct monthly climatologies of

245surface ocean $pCO_2$, was assessed by comparing collocated reconstructed $pCO_2$ and corresponding values

246from Takahashi et al. (2009). The global climatology was reconstructed with a satisfying accuracy during

247step 1 with a RMSE of 0.17 µatm and $r^2$ of 0.93. Model output of step 2 was assessed by K-fold cross

248validation as presented before: K=4 different subsamplings of independent data were drawn from the data

249set and the network was run 5 times on each subsampling. From these 20 results the best one was chosen

250based on RMSE, $r^2$ and bias. The combination of the four best model output was used for the statistical

251analysis summarized in Table 1. Metrics were computed over the full period (2001-2016) and with

252reference to SOCAT observations (independent data only). At the global scale, the analysis yielded a RMSE

253of ~17.97 µatm, while the absolute bias was 11.52 µatm and $r^2$ 0.76. These results are comparable to those

254obtained by Landschützer et al. (2013) for the assessment of a surface ocean $pCO_2$ reconstruction based on

255an alternative neural network based approach. The RMSE between SOCAT data and the climatology of

256$pCO_2$ from Takahashi et al. (2009) equals 41.87 µatm, larger than erros computed for the regional

257comparison between FFNN and SOCAT (Table 1).

259Figure 2 (a) shows the time mean difference between the estimated $pCO_2$ and $pCO_2$ from SOCAT v5 data

260used for validation $mean_t \left( pCO_{2,i,j,FFNN} - pCO_{2,i,j,SOCAT} \right)$ . Large differences occurred at high

261latitudes, in equatorial regions, along the Gulf Stream and Kuroshio currents – the regions with strong

262horizontal gradient of $pCO_2$. Moreover the standard deviation of residuals (Figure 2 (b)) in these regions

263was larger indicating that the model fails to accurately reproduce the temporal variability. The reduced skill

264of the model in these regions reflects the poor data coverage along with a strong seasonal variability (e.g.

265Southern Ocean) and/or high kinetic energy (e.g. Southern Ocean, Kuroshio and Gulf Stream currents)

266(Fig. 1 (a)). At the scale of ocean regions, (Table 1) the largest RMSE and bias were computed for the

267Pacific Subpolar ocean (RMSE = 34.77 µatm, biais = 23.12 µatm), while the lowest correlation coefficient

268was obtained for the equatorial Atlantic ocean ($r^2$ = 0.57). These low scores directly reflect low data density

269and are to be contrasted with those obtained over regions with better data coverage (e.g. Subtropical

270Pacific: RMSE = 15.86 µatm, biais = 9.9 µatm, $r^2$ = 0.77 or Subpolar Atlantic: RMSE = 22.99 µatm, biais =

27115.04 µatm, $r^2$ = 0.76). Despite large time mean differences computed over the eastern Equatorial Pacific,

272scores are satisfying at the regional scale indicating error compensation by improved scores over the

273western basin. Scores are low in the Southern hemisphere (Table 1) and time mean differences are large

274(Fig. 2 (a)) reflecting sparse data coverage (Fig. 1 (a)).

2763.2. Qualification.

277This section presents the assessment of the final time series of reconstructed surface ocean $pCO_2$. The time

278series was computed using the best monthly models as described in section 2.2, as well as 100% of data for

278learning, evaluation and validation.

279Results of the FFNN-LSCE mapping model were compared to three published mapping methods which
280participated in the "Surface Ocean pCO2 Mapping Intercomparison" (SOCOM) exercise presented in
281Rödenbeck et al. (2015) (http://www.bgc-jena.mpg.de/SOCOM/). These methods are: (1) Jena-MLS
282(Rödenbeck et al., 2014), a statistical interpolation scheme (data-driven mixed-layer scheme; principal
283drivers: ocean-internal carbon sources/sinks, SST, wind speed, mixed-layer depth climatology, alkalinity
284climatology); (2) JMA-MLR (Iida et al., 2015), based on multi-linear regressions with SST, SSS and Chl *a*
285as independent variables, and (3) ETH-SOMFFN (Landschützer et al., 2014), a combined two-step neural

[revised manuscript text omitted]

411Figure 8 shows the linear trends of sea-air $CO_2$ fluxes for FFNN-LSCE (a), Jena-MLS13 (b), ETH-

412SOMFFN (c) and JMA-MLR (d). A total negative trend was computed for all models, albeit with large

413regional contrasts, and FFNN-LSCE  within the range: Jena-MLS13, -0.0028 PgC/yr; FFNN-LSCE,

414-0.0032 PgC/yr; JMA-MLR, -0.0037 PgC/yr; ETH-SOMFFN, -0.0059 PgC/yr. FFNN-LSCE computed

415negative trends over most of the Atlantic basin, Indian Ocean and South of 40ºS, which contrasts with

416decreasing fluxes over the Pacific and locally in the Antarctic Circumpolar  At first order this broad

417regional pattern is found in all models. Regional maxima and minima are, however, more pronounced in

418Jena-MLS13 (Fig. 8 (b)) and ETH-SOMFFN (Fig. 8 (c)), while a patchy distribution at sub-basin scale is

diagnosed for JMA-MLR.

The agreement in sign of computed linear trends from four models is presented in Fig. 9. Over most of the
ocean, all four models show very close sea-air $CO_2$ tendency. In the Indian Ocean (biome 14), on the other
hand a positive trend was computed for JMA-MLR (0.0004 PgC/yr) while the three other models present a
negative trend.  These differences between models were also found in the Pacific Ocean, especially the
Southern Pacific. In the Eastern Equatorial Pacific region (biome 6) a total negative trend equal to
$-4.03 \times 10^{-5}$ PgC/yr was computed for ETH-SOMFFN, which contrasts with positive trends suggested by
FFNN-LSCE ($6.68 \times 10^{-5}$ Pg/C/yr) and Jena-MLS13 ($3 \times 10^{-4}$ PgC/yr). All models reproduced a maximum in
the southern part of biome 6 but they disagree about its amplitude and spatial distribution. Almost
everywhere over the Atlantic Ocean the mapping methods produced the same sign of linear trend (Fig. 9).
Only in the eastern part of the subtropical North Atlantic Jena-MLS13 gave a positive linear trend of $fCO_2$
(Fig. 8 (b)).

According to FFNN-LSCE, the global ocean took up in average 1.55 PgC/yr between 2001-2015.
consistent with results from the other three models (Table 3) (see Table S1 for estimations per
biomes). The spread between individual models falls in the range of the error reported in Landschützer et
al. (2016), ±0.4-0.6 PgC/yr. Per biome, estimates of $CO_2$ sea-air fluxes provided by FFNN-LSCE are
similarly in good agreement with those derived from the other models.

**4. Summary and conclusion.**

We proposed a new model for the reconstruction of monthly surface ocean $pCO_2$. The model is applied
globally and allows a seamless reconstruction without introducing boundaries between the ocean basins or
Our model relies on a two-step approach based on Feed-Forward Neural Networks (FFNN-LSCE).
The first step corresponds to the reconstruction of a monthly $pCO_2$ climatology.  allows us to keep the
output of the FFNN close to the observed values in the region with the poor data cover. Moreover, it allows
to include a potential change in seasonal cycle in response to climate change from drivers to carbon cycle
variables. At the second step $pCO_2$ anomalies are reconstructed according to climatology from the first step.
The model was applied over the period 2001-2016. Validation with independent data at global scale
indicated an accuracy of 17.57 µatm, $r^2$ of ~0.76 and an absolute bias of 11.52 µatm. In order to assess the
model further, it was compared to three different mapping models: ETH-SOMFFN (self-organizing maps +
neural network), Jena-MLS13 (statistical interpolation), JMA-MLR (linear regression) (Rödenbeck et al.,
2015). Network qualification followed the protocol and diagnostics proposed in Rödenbeck et al. (2015).
Reconstructed surface ocean $pCO_2$ distributions were in good agreement with other models and
observations. The seasonal variability was reproduced satisfyingly by FFNN-LSCE, the yearly $pCO_2$

mismatch varied around zero, and relative IAV mismatch was 7.55%. FFNN-LSCE proved skillful in reproducing the interannual variability of surface ocean $pCO_2$ over the Eastern Equatorial Pacific in response to ENSO. Reductions in surface ocean $pCO_2$ during El Niño events were well reproduced. The comparison between reconstructed and observed $pCO_2$ values yielded a RMSE of 15.73 µatm, $r^2$ of 0.79

and an absolute bias of 10.33 µatm over the Equatorial Pacific. The relative IAV misfit in this region was

~17%. Despite an overall good agreement between models, important differences still exists at the regional scale, especially in the Southern hemisphere and in particular, the Southern Pacific and the Indian Ocean.

These regions suffer from poor data-coverage. Large regional uncertainties in reconstructed surface ocean

$pCO_2$ and sea-air $CO_2$ fluxes have a strong influence on global estimates of $CO_2$ fluxes and trends.

**Code and data availability.**

Python code for $pCO_2$ climatology reconstruction, $1^{st}$ step of FFNN-LSCE model:

https://files.lsce.ipsl.fr/public.php?service=files&t=016351132f69db55f1e6eda948665237

Python code for reconstruction of $pCO_2$ anomalies, $2^{nd}$ step of FFNN-LSCE model:

https://files.lsce.ipsl.fr/public.php?service=files&t=9304199cf79efd688837e891383287c3

Data of FFNN-LSCE $pCO_2$ are available on request: anna.sommer.lab@gmail.com

**Author contribution.**

[revised manuscript text omitted]
 January for period 2001-2016; (c) - all months December-January-February for period 2001-2016.**

[Figure]

[Figure]

[Figure]

**Figure 2: Time mean differences (µatm) (a) between monthly FFNN-LSCE pCO₂ and SOCAT pCO₂ data used for evaluation of the model over the period 2001-2016 and its std (b).**

[Figure]

**Figure 3: Map of biomes (after Rodenbeck et al. (2015); and Fay and McKinley (2014)) used for comparison. See table 2 for biome names.**

[Figure]

[Figure]

**Figure 4: Global oceanic pCO$_2$: black - FFNN-LSCE, blue - JMA, brown - Jena, green - ETH-SOMFFN; (a) - monthly time series averaged over the glob, (b) - 12-month running mean averaged over the glob, (c) - yearly pCO$_2$ mismatch (difference of mapping methods and SOCAT data).**

[Figure]

[Figure]

**Figure 5: East Pacific Equatorial (biome 6) (left) and North Atlantic Subtropical Permanently Stratified (biome 11) (right) oceanic pCO₂: black – FFNN, blue – JMA, brown – Jena, green – ETH-SOMFFN; (a), (b) – monthly time series averaged over biome; (c), (d) – 12-month running mean averaged over biome; (e), (f) – yearly pCO₂ mismatch (difference of mapping methods and SOCAT data).**

[Figure]

[Figure]

**Figure 6: (a) – Interannual sea-air CO₂ flux (12-month running mean) in the global ocean; (b) – amplitude of interannual CO₂ flux plotted against the relative IAV mismatch amplitude. The weighted mean is given as a horizontal line.**

[Figure]

[Figure]

**Figure 7: East Pacific Equatorial (biome 6) (left) and North Atlantic Subtropical Permanently Stratified (biome 11) (right): (a), (b) – Interannual sea-air CO₂ flux (12-month running mean) in the global ocean; (c), (d) – amplitude of interannual CO₂ flux plotted against the relative IAV mismatch amplitude. The weighted mean is given as a horizontal line.**

[Figure]

[Figure]

[Figure]

**Figure 8: Linear trend of fCO$_2$ for common period 2001-2015: (a) – FFNN-LSCE; (b) – Jena-MLS13; (c) – ETH-SOMFFN; (d) – JMA-MLR.**

[Figure]

[Figure]

**Figure 9: Agreement between four mapping methods in their linear trend of sea-air CO₂ flux. Color-bar represents the number of products that have the same sign of linear trend.**

Table 1: Statistical validation of FFNN-LSCE. Comparison between reconstructed surface ocean pCO₂ and pCO₂ values from SOCAT v5 data base not used in the training algorithm for the period 2001-2016 over the global ocean (except for regions with ice-cover) and for large oceanographic regions. In round brackets:

[Figure]

727number of measurements per region

| Model | Latitude boundaries | RMS (µatm) | r² | Bias (µatm) |
|---|---|---|---|---|
| FFNN Global | | 17.97 | 0.76 | 11.52 |
| Arctic (150) | 76ºN to 90ºN | 22.05 | 0.54 | 17.1 |
| Atlantic Subpolar (21903) | 49ºN to 76ºN | 22.99 | 0.76 | 15.04 |
| Pacific Subpolar (4529) | 49ºN to 76ºN | 34.77 | 0.65 | 23.12 |
| Atlantic Subtropical (41331) | 18ºN to 49ºN | 17.28 | 0.69 | 11.27 |
| Pacific Subtropical (41867) | 18ºN to 49ºN | 15.86 | 0.77 | 9.9 |
| Atlantic Equatorial (7300) | 18ºS to 18ºN | 17.27 | 0.57 | 11.44 |
| Pacific Equatorial (27092) | 18ºS to 18ºN | 15.73 | 0.79 | 10.33 |
| South Atlantic (3002) | 44ºS to 18ºS | 17.81 | 0.63 | 12.28 |
| South Pacific (12934) | 44ºS to 18ºS | 13.52 | 0.63 | 9.36 |
| Indian Ocean (2871) | 44S to 30N | 17.25 | 0.62 | 11.6 |
| Southern Ocean (16334) | 90ºS to 44ºS | 17.4 | 0.58 | 11.92 |

729Table 2: Biomes from Fay and McKinley (2014) used for time series comparison (Fig. 3)

| Number | Name |
|---|---|
| 1 | (Omitted) North Pacific Ice |
| 2 | North Pacific Subpolar Seasonally Stratified |
| 3 | North Pacific Subtropical Seasonally Stratified |
| 4 | North Pacific Subtropical Permanently Stratified |
| 5 | West Pacific Equatorial |
| 6 | East Pacific Equatorial |
| 7 | South Pacific Subtropical Permanently Stratified |
| 8 | (Omitted) North Atlantic Ice |
| 9 | North Atlantic Subpolar Seasonally Stratified |
| 10 | North Atlantic Subtropical Seasonally Stratified |
| 11 | North Atlantic Subtropical Permanently Stratified |
| 12 | Atlantic Equatorial |
| 13 | South Atlantic Subtropical Permanently Stratified |
| 14 | Indian Ocean Subtropical Permanently Stratified |

[Figure]

[Figure]

[Figure]

| 15 | Southern Ocean Subtropical Seasonally Stratified |
| 16 | Southern Ocean Subpolar Seasonally Stratified |
| 17 | Southern Ocean Ice |

Table 3: Mean of sea-air $CO_2$ flux (PgC/yr) over the Global Ocean and per regions for period in common
(2001-2015). Averages over the period 2001-2009 are presented between brackets. The last column
presents a comparison to best estimates from Schuster et al. (2013) for the Atlantic Ocean (1990 – 2009).

| Region | Latitude boundaries | FFNN-LSCE | ETH-SOMFFN | Jena-MLS13 | JMA-MLR | Schuster et al. (2013), 1990-2009 |
|---|---|---|---|---|---|---|
| Global | | -1.55 (-1.44) | -1.67 (-1.47) | -1.55 (-1.41) | -1.74 (-1.62) | --- |
| Arctic | 76ºN to 90ºN | -0.001 | -0.001 | -0.001 | -0.001 | -0.12±0.06 |
| Atlantic Subpolar | 49ºN to 76ºN | -0.15 (-0.15) | -0.14 (-0.12) | -0.15 (-0.15) | -0.16 (-0.15) | -0.21±0.06 |
| Pacific Subpolar | 49ºN to 76ºN | -0.003 (-0.005) | -0.009 (-0.004) | -0.006 (-0.004) | -0.027 (-0.021) | --- |
| Atlantic Subtropical | 18ºN to 49ºN | -0.21 (-0.19) | -0.21 (-0.19) | -0.2 (-0.18) | -0.21 (-0.2) | -0.26±0.06 |
| Pacific Subtropical | 18ºN to 49ºN | -0.45 (-0.46) | -0.49 (-0.48) | -0.47 (-0.46) | -0.49 (-0.47) | --- |
| Atlantic Equatorial | 18ºS to 18ºN | 0.085 (0.09) | 0.085 (0.095) | 0.08 (0.082) | 0.1 (0.11) | 0.12±0.04 |
| Pacific Equatorial | 18ºS to 18ºN | 0.42 (0.41) | 0.4 (0.4) | 0.44 (0.42) | 0.38 (0.37) | --- |
| South Atlantic | 44ºS to 18ºS | -0.17 (-0.16) | -0.18 (-0.16) | -0.18 (-0.17) | -0.23 (-0.22) | -0.14±0.04 |
| South Pacific | 44ºS to 18ºS | -0.33 (-0.34) | -0.4 (-0.39) | -0.35 (-0.34) | -0.49 (-0.47) | --- |
| Indian Ocean | 44S to 30N | -0.25 (-0.2) | -0.32 (-0.29) | -0.27 (-0.26) | -0.27 (-0.29) | --- |
| South Ocean | 90ºS to 44ºS | -0.38 | -0.29 | -0.36 | -0.26 | --- |

---

## Author Comment (AC1) · 13 Mar 2019

We would like to thank Astrid Kerkweg for her comments about the journal requirements.

We added in the title of the manuscript a number of model version. The program codes of model will be added in supplementary materials. The model data will be accessible at CMEMS (Copernicus Marine environment monitoring service) website. We will provide a corresponding information in the revised version.

[Figure]

2018.

---

## Author Comment (AC2) · 13 Mar 2019

We thank Dr Christian Rödenbeck for his comments and suggestions. The manuscript was revised to address each point.

Christian Rödenbeck: The authors present a method to interpolate temporally and spatially discrete surface- ocean pCO2 observations into a gridded field, using non-linear regression against var- ious environmental drivers. Compared to similar methods, two specific features are (a) estimating seasonality separately based on data also outside the calculation period, thereby improving its data constraint, and (b) the addition of SSH to the set of predictors. This is an interesting contribution to the ensemble of existing interpolation methods, enlarging the range of plausible outcomes. The manuscript nicely describes the details of the method, and presents evaluation metrics, also in the context of other methods from the SOCOM ensemble. I clearly recommend to publish this manuscript. Below some suggestions mainly to further improve clarity at a few places.

Authors: Thank you very much for your positive evaluation. We provide hereafter a detailed point-by-point reply.

Christian Rödenbeck: The method is very similar to CARBONES-NN (not published in the peer-reviewed literature but described in the SOCOM paper RoÌLdenbeck et al., 2015), developed at LSCE as well. I therefore assume that the presented method builds on and supersedes CARBONES-NN. Is this correct? If so, this is an interesting piece of information to users of the SOCOM ensemble and should be mentioned in this paper. Further, I would find it fair to give credit to the authors of CARBONES-NN (to my knowledge, Abdou Kane and Philippe Peylin).

Authors: The present study (by Denvil-Sommer et al.) reflects our current vision of this particular estimation problem. It is part of a long-standing activity at LSCE with focus on the surface ocean carbon system. Abdou Kane and Philippe Peylin's work, the development of the CARBONES-NN model, is an earlier example. The model has been largely redesigned to turn into the one presented in the current submission. Hence, the new model presented in this paper replaces the former one. Philippe Peylin is of course fully aware of our latest efforts, but his work has now other priorities.

Christian Rödenbeck: You mention the inclusion of SSH, but do not discuss it. How does it influence, and possibly improve, the results?

Authors: First tests suggested that the inclusion of SSH does not significantly improve the accuracy of reconstructed pCO2 at global scale. We added a corresponding sentence in the manuscript. At basin and regional scale, however, adding SSH improves the spatial pattern of reconstructed pCO2 and the accuracy of our method. The full assessment of SSH will be part of a follow-up study.

C.R.: How much do the data outside the period improve the estimates of seasonality from the 1st step?

A.: The climatology by Takahashi et al. (2009) is based on data obtained from the early 1970s to 2007. Coastal pCO2 measurements and values obtained in the Eastern Equatorial Pacific are excluded. It represents average open ocean pCO2 conditions and is centred on the year 2000. A way to estimate the contribution of data outside the reconstruction period to estimates of seasonality would consist in computing the climatology over the period covered by the reconstruction only. A sensitivity analysis could also be carried out subsampling a numerical model. However, since Takahashi et al. excluded years with strong anomalies, we expect the influence of data outside of the reconstruction period to be small.

C.R.: Couldn't step 1 also be fitted against the SOCAT data (folded into a climatological year as done by Takahashi et al., 2009) as well, rather than against the already-interpolated climatology? This would also bring in the data from the most recent years not yet in the Takahaski et al. (2009) climatology.

A.: This could indeed by an alternative approach to step 1, which could be tested in the future. The use of an interpolated climatology allows however to benefit from a full data coverage (a larger data set, as stated in line 78), which would not be the case with SOCAT.

C.R.: Title: While there is no question that the authors are fully free to choose a name for their method, it seems that they intended to use a "SOCOM-style" name. Therefore I'd like to point out that all names in the SOCOM paper are of the form INSTITUTION-METHOD, not the other way round.

A.: We followed the reviewer's suggestion and changed the name of the model to LSCE-FFNN.

C.R.: Line 78: "On a larger data set" - please clarify what this means

A.: The use of a gridded climatology during the first step allows benefiting from a global and continuous data coverage. It provides more data for training and validation. It was clarified in the text.

C.R.: Lines 79-81 and 446: Unclear statement - isn't it the 2nd step (not the 1st) which takes care of changes?

A.: The 1st step reconstructs the climatological seasonal variability. The 2d step adds interannual variability to the signal if present. It contributes information on anomalies with respect to the climatological state.

C.R.: Line 114: If you took surface pressure from the s76_v4.1 run, it is in fact from NCEP (Kalnay et al.). This should be mentioned."

A.: It was modified and mentioned in the text.

C.R.: Lines 141, 142, 191: The "1" and "2" are awkward, and should rather be part of the subscript.

A.: It was modified.

C.R.: Line 174: Isn't the 1st step using climatologies of the driver variables? Please clarify. Also, is it correct that the 1st step uses un-normalized drivers (rather than $SST_n$ etc.)?

A.: Yes, driver climatologies are used during step 1. All data have to be normalized for both steps of the model. It was clarified in the text.

C.R.: Line 191: I assume that the "n" in the subscript to pCO2 is not correct, is it?

A.: It is correct. "n" stands for "normalized". All variables are normalized prior to their use in the model. It was clarified.

C.R.: Line 197: Maybe say "for each climatological month".

A.: Corrected.

C.R.: Line 209: Does "chosen automatically" mean randomly? Please clarify.

A.: Yes, it is randomly. It was modified in the text.

C.R.: Line 212: What does "final model" mean? Is it the model of step 2?

A.: Yes it is step 2. We have clarified it.

C.R.: Line 236: "(4)" should probably be "(5)".

A.: Corrected.

C.R.: Line 238: Is it from run s76_v4.1 as well? Please mention.

A.: Yes, it is. We have clarified it.

C.R.: Line 244: Isn't "did not participate" in contradiction to Sect 2.2 b) which says "100% used for training"?

A.: We distinguish between the development of the model presented in section 2.2 a) and its application to pCO2 reconstruction in section 2.2 b). It is stated in the manuscript as follows: "The previous section presented the development of a FFNN model for the reconstruction of global surface ocean pCO2, and the estimation of its accuracy". During development, the accuracy of the method was evaluated based on 25% of data that did not participate in training. We added a sentence for clarification.

C.R.: Line 283: Though the Jena scheme indeed uses SST etc in its parameterizations, the sentence can be misunderstood as meaning regression drivers. Maybe just remove the list of drivers here.

A.: We prefer to provide a list of variables used. We have added that these variables are used in its parameterizations.

C.R.: Line 285: I feel "combined" is confusing and can be removed.

A.: We agree and removed it.

C.R.: Line 286: Please mention whether you used the same versions of these methods as used in the SOCOM paper, or whether you used updated versions. If updated, the respective version IDs should be given.

A.: We have added the number of versions used.

C.R.: Line 286-287: The sentence "Qualification...(2015)." is repeated in the next paragraph and may be deleted here. I feel the next sentence "The time series..." should start a new paragraph.

A.: We agree, the sentence was deleted.

C.R.: Line 392: I guess "small IAV" should be "small relative IAV mismatch", isn't it?

A.: It is. It was corrected.

C.R.: Line 394, 296, 403, 455: The figures after the decimal point are certainly meaningless and should be removed.

A.: We agree, it was corrected.

C.R.: Lines 412, 425: It is unclear to me what "total negative trend" means.

A.: It is a linear trend averaged over the globe. It was clarified in the text.

C.R.: Line 413 and below: "PgC/yr" is the unit of a flux, not of a flux trend. Please clarify the unit (it should be something like "PgC/yr/yr" or "PgC/yr/decade").

A.: Yes, indeed, it is PgC/yr/yr. This has been corrected.

C.R.: Lines 421ff: Are trends like 0.0004PgC/yr/yr statistically significant given the IAV? If not, it is not actually appropriate to call them "positive".

A.: Yes, t-test showed that it is significant (p-val is 0.05).

C.R.: Figures: In my print-out, the annotations and labels of most figures are much too small to be readable.

A.: The figures will be changed in the revised version.

C.R.: Fig 9, color bar: In this figure, the numbers should be within the colors, not between them, otherwise the meaning is ambigious.

A.: Done.

C.R.: Fig S7 caption: "according" probably means "agreement"?

A.: Corrected.

All typos were corrected.
* * *

---

## Author Comment (AC3) · 13 Mar 2019

We would like to thank Luke Gregor for his positive and helpful review. We have answered his comments below.

Luke Gregor: The study introduces a new method that calculates pCO2 as climatology + anomaly. I think that this is a novel approach and is definitely relevant to the community. The authors do a good job of explaining the method and comparing it with past studies. I really like the approach and the method seems to perform well relative to the other methods in the SOCOM ensemble. I enjoyed reading the manuscript; however, there are many typos scattered throughout the document. I have noted most of these

with track changes in the PDF document using Adobe Reader (attached as "supplement"). Below are some general comments, then repeated mistakes that can be fixed with find/replace.

Authors: Thank you for the positive assessment of our study. We apologize for the typos.

L.G.: The authors limit their study from 2001 to 2016. I realise that data is sparse before 2001, but it would be interesting to see how the model fares. It is useful for the community to know if this method (and in general the climatology-regression approach) can predict in data-sparse periods. It seems to be good at predicting in data-sparse regions.

A.: Thank you for your suggestion. We agree that the extension of the time series is a priority for future developments.

L.G.: It is really good that the authors use quasi-regular spacing to define the train-test validation splits, but I would like to know more about how they defined these splitting points. On this point, I also think it would be useful to have a measure of how much a model overfits. The authors could compare the test RMSE with the RMSE of the model trained with 100% of the data. If the latter is much smaller than the 50:25:25 splits, then there is overfitting. This is important for when the method is compared with other gap-filling methods.

A.: To be used by the FFNN algorithm data have to be formatted as a list "latitude by latitude". Data are read "line by line" with each line corresponding to a latitude. Figure S1 presents the method of selection that was applied. This approach ensures that information from almost all regions and months are used for training. The RMSE for the case when 100% of data were used for training amounts 14.8 $\mu$atm, only slightly smaller than the one computed for cross-validation (17.97 $\mu$atm). We added this information to the manuscript.

L.G.: Regarding the figures, I struggle to see the difference between the brown and the black lines as I'm slightly colourblind. I would like to see another colour for the Jena-MLS method.

A.: Colour was changed to red.

L.G.: "Methods based on ANN are able to represent the large class of pCO2 -driver relationships, but they are sensitive to the number of data used in the training algorithm and can generate artificial variability in regions with sparse data coverage.". - I find the first part of the sentence a bit unclear. Not sure what is meant by "large class of pCO2-driver relationships". The second part of the sentence requires a reference.

A.: "Large class of pCO2-driver relationships" refers to a large variety of combinations of drivers/predictors. The first part of the sentence was modified to "Methods based on ANN are able to represent the relationship between pCO2 and a variety of predictor combinations (e.g. pCO2=f(SSS,SST,SSH) or pCO2=f(SSS,SST,xCO2,CHL,MLD)). A reference was added.

L.G.: "The model is easily applied to the global ocean without any boundaries between the ocean basins or regions". - This amazes me! Good results with one domain. A great advantage to the method!

A.: Thank you!

L.G.: "Based on Rodgers et al. (2009) who reported a strong correlation between natural variations in dissolved inorganic carbon (DIC) and sea surface height (SSH), SSH was added as a new driver to this list". - I like the addition of SSH to the list, but then why not extend the predictions to 1993 (SSH limiting) or 1998 (Chl-a). I agree with Reviewer 1 that this can be discussed more.

A.: In this study, the priority was given to data coverage for the development of the model. The period retained (2001-2016) represents 77% of the SOCAT dataset. The extension of the period of reconstruction will be considered in future studies.

[Figure]

L.G.: I would have hoped that the climatology approach would be better at estimating pCO2 for periods and regions where data is sparse.

A.: It is difficult to assess the skill of the model over regions with sparse data coverage based on observations only. An alternative approach would consist in subsampling a numerical model with the temporal and spatial coverage of real observations. These pseudo-data will be used for reconstructing surface ocean pCO2. The comparison between reconstructed and modelled pCO2 distributions will provide an upper estimate of the accuracy.

L.G.: "Monthly global observed physics reprocessed products distributed through the Copernicus Marine Environment Monitoring Service (CMEMS) (0.25ox0.25o) (http://marine.copernicus.eu/services-portfolio/access-to-products/?option=com_csw&view=details&product_id=MULTIOBS_GLO_PHY_REP_015_002) were used for SSS, SST and SSH". - ARMOR3D L4? Reference: Guinehut et al 2012?

A.: Yes, it is the ARMOR3D L4 data. We added the reference Guinehut et al. (2012).

L.G.: "For MLD, daily data from the "Estimating the Circulation and Climate of the Ocean" (ECCO2) project Phase II, at 0.25ox0.25o resolution (Menemenlis et al., 2008) were used". - Please add the type of data product.

A.: The product name is ECCO2 and the release name is Cube 92. The information was added to the manuscript.

L.G.: Lignes 140-142 - Converted to radians first.

A.: Yes, of course. The information was added.

L.G.: "There is a well-known empirical rule advising to have a factor of 10 between number of patterns (data) and number of connections, or weights to adjust". - Please add a reference. COMMENT: Landschutzer 2013 used 30. Are there any signs of overfitting to the training data? i.e. is the training RMSE score much lower than the test RMSE score?

A.: Landschutzer et al. (2013) based their choice on Amari et al. (1997). A factor of 30 allows avoiding cross-validation. However, we believe that a problem such as the reconstruction of the time and spatial distribution of surface ocean pCO2 requires cross-validation. In line with Amari et al. (1997), we use a factor of 10 that necessitates a cross-validation to avoid overfitting.

L.G.: "Validation and evaluation data sets were chosen quasi-regularly in space and time to take into account all regions and seasonal variability". - This is good, but I would like to know more about how the quasi-regular sampling was done.

A.: As mentioned above, to be used by the FFNN algorithm data have to be formatted as a list "latitude by latitude". Data are read "line by line" with one line per latitude. This approach ensures that information from almost all regions and months are used for training.

L.G.: "In order to improve the accuracy of the reconstruction, the model was applied separately for each month". - So in effect, 12 models were trained? This is mentioned later in the text. Move that point here.

A.: The corresponding sentence was added.

L.G.: "We have developed a FFNN model with 5 layers (3 hidden layers)". - Sentence is unclear in this context. (Put earlier).

A.: We placed this sentence earlier in the manuscript.

L.G.: "There were thus 12 models sharing a common architecture but trained on different data". - This could move to the first step to clarify the point a bit earlier. I have noted where this point could be moved to.

A.: We have added a sentence to the description of the first step. We kept the sentence as it corresponds to the 2nd step of the model and we feel that it is needed for clarification.

L.G.: "At this step, in order to increase the amount of data during training and to introduce information on the seasonal cycle, the model was trained using as a target pCO2 data from the month in question as well as those from the previous and following month during the entire period 2001-2016". - My feeling is that having monthly models would really reduce the number of data in the Southern Ocean during winter. Really very little data. I have made a comment about this for Figure 1.

A.: We agree that the Southern Ocean is particularly under-sampled with a bias towards summer months. However, the use of a single model for the reconstruction of surface ocean pCO2 over the full period will not significantly improve the accuracy in the Southern Ocean, but it will reduce it over the global ocean.

L.G.: "Figures 1 (b) and 1 (c) show an example of data distribution for the sole months of January over the period 2001-2016 (Fig. 1 (b)) and for the three months time-window December-January-February 2001-2016 used in the training algorithm of the January FFNN model (Fig. 1 (c))". - I would like to see this for July rather - This would really highlight the missing data for the Southern Ocean. Also, the figure is very small. I would prefer if the subplots were vertically stacked (single column).

A.: The aim of these figures is to show a possible distribution of data and an impact of data from two neighboring months. We did not want to focus on a particular month and discuss associated problems. Figures were vertically stacked.

L.G.: "Each sampling was tested..." - "sampling fold" would be a lot clearer.

A.: We changed the text accordingly.

L.G.: "...different initial values that are chosen automatically". - how are these chosen automatically? There must be an initialisation or a grid of hyper-parameters passed to the model. Please clarify.

A.: The values are chosen randomly.

L.G.: "Note that a higher number of parameters did not show a significant improvement of accuracy (not shown)". - Remove (not shown). Makes me want to see this. COMMENT: This is good. shows that over-training is being minimised.

A.: Done.

L.G.: "In order to provide the final output, the selected FFNN architecture is trained on all available data: 100% of data for training, 100% for evaluation and 100% for validation". - I would like to see RMSE from validation data pre- and post-training, but only necessary in the supplementary materials. The authors can then state that their method is or is not prone to overfitting. This would show if there is overtraining.

A.: Thank you for the suggestion. We agree that it would be interesting to present these values to provide additional information on the model development. We have noted it for the next model release. The RMSE for the case of 100% of training data is 14.8 uatm. The information was added to the manuscript.

L.G.: Ligne 225: FFNN-LSCE

A.: Corrected.

L.G.: ". The global climatology was reconstructed with a satisfying accuracy during step 1 with a RMSE of 0.17 $\mu$atm and r2 of 0.93". - This is very low! might be worth-while commenting on. This shows that pCO2 deviation from the climatology contributes nearly all the error in estimates!

A.: The high scores during step 1 underline the capacity of our FFNN approach to reconstruct the surface ocean pCO2 when there is a good data coverage. However, it is hard to conclude whether the 1st or the 2nd step has more or less contribution to total model accuracy. As you and Christian Rödenbeck mentioned, the estimation of the climatology during step 1 includes data outside of the reconstruction period, which might influence final results. This needs more investigation using numerical models, for example. However, we expect that the influence of the data outside of the period will not significantly modify our results.

L.G.: Ligne 253: absolute bias - Rather refer to this as the mean absolute error and I would also like to see the bias.

A.: We have found that total bias is difficult to explain as it varies a lot over the global ocean and its small value does not mean that the biases are small everywhere, often it results from compensation. We added a table presenting the bias to the Supplementary material. "Absolute bias" was replaced by "mean absolute error" (MAE).

L.G.: "The RMSE between SOCAT data and the climatology of pCO2 from Takahashi et al. (2009) equals 41.87 $\mu$atm, larger than errors computed for the regional comparison between FFNN and SOCAT (Table 1)". - Is this given as a normalisation/comparison - I think this is quite useful.

A.: Indeed, this is given as a comparison. The comparison stresses the skill of the model to reconstruct interannual anomalies. An error between reconstructed pCO2 and SOCAT larger than between the climatology by Takahashi et al. (2009) and SOCAT would indicate that the model failed the capture the interannual variability.

L.G.: Ligne 269: "...Subtropical Pacific..." - assuming NH, but better to clarify.

A.: Done.

L.G.: "Despite large time mean differences computed over the eastern Equatorial Pacific, scores are satisfying at the regional scale indicating error compensation by improved scores over the western basin". - I'm glad this is spoken about. Please add a number to explain "satisfying".

A.: Done.

L.G.: Ligne 273: "...South hemisphere..." - Southern Hemisphere, each should be capitalised.

A.: Done.

All typos and mistakes were corrected.